# Cancer–immune coevolution dictated by antigenic mutation accumulation

**Long Wang[1,2], Christo Morison[2], Weini Huang[1,2]***

[1]Group of Theoretical Biology, Innovation Center for Evolutionary Synthetic Biology School of Life Sciences, Sun Yat-sen University, Guangzhou, China; [2]School of Mathematical Sciences, Queen Mary University of London, London, United Kingdom

## eLife Assessment

This **important** work presents a stochastic branching process model of tumour-immune coevolution, incorporating stochastic antigenic mutation accumulation and escape within the cancer cell population. They then used this model to investigate how tumour-immune interactions influence tumour outcome and the summary statistics of sequencing data of bulk and single-cell sequencing of a tumour. The evidence is **compelling** and the work will be of interest to cancer-immune biology fields.

**\*For correspondence:**
weini.huang@qmul.ac.uk

**Abstract** The immune system is one of the first lines of defence against cancer. When effector cells attempt to suppress tumour, cancer cells can evolve methods of escape or inhibition. Knowledge of this coevolutionary system can help to understand tumour–immune dynamics both during tumourigenesis and during immunotherapy treatments. Here, we present an individual-based model of mutation accumulation, where random mutations in cancer cells trigger specialised immune responses. Unlike previous research, we explicitly model interactions between cancer and effector cells and incorporate stochastic effects, which are important for the expansion and extinction of small populations. We find that the parameters governing interactions between the cancer and effector cells induce different outcomes of tumour progress, such as suppression and evasion. While it is hard to measure the cancer–immune dynamics directly, genetic information of the cancer may indicate the presence of such interactions. Our model demonstrates signatures of selection in sequencing-derived summary statistics, such as the single-cell mutational burden distribution. Thus, bulk and single-cell sequencing may provide information about the coevolutionary dynamics.

## Introduction

The immune system does not only protect our body from infectious diseases caused by various pathogens, but also gives the first response against threats emerging within the body, such as cancer. Immune cells try to identify and eliminate tumour cells, which may express antigens not found on normal cells. Meanwhile, tumour cells attempt to hide or evade from immune surveillance (*Hanahan, 2022*). From an ecological perspective, this antagonistic relationship gives rise to complex dynamics between a tumour and its microenvironment (*Dujon et al., 2021*; *Kareva et al., 2021*). On an evolutionary level, these two cell types, though belonging to the same organism, coevolve, each adapting to genotypic and phenotypic changes in the other population (*George and Levine, 2020*). We are interested in the eco-evolutionary dynamics of genotype-specific interactions between cancer and immune cells arising from a continuous introduction of new antigens in the context of immune surveillance and escape.

**eLife digest** The human body is made up of around 36 trillion cells and 200 different types of cells, each with a specialised role. For example, immune cells are crucial for fighting infections. They also act as the body's first line of defence against internal threats, such as cancer.

Cells have intricate systems to control how often and how much they divide, ensuring a fresh supply of cells. When some of these signals are faulty or missing, cells can start to grow and multiply uncontrollably. This unchecked growth can form tumours.

Some immune cells can recognise cancer cells by certain proteins on their surface and mark them for destruction. This process is thought to eliminate many potential tumours before they become dangerous. This way, early cancer can be held in check for years.

However, cancer cells can evolve genetic changes that help them evade the immune system. In response, immune cells adapt to find new ways to identify these mutations. This creates an evolutionary arms race, with each side developing new strategies to outsmart the other. Eventually, cancer cells may acquire enough changes to grow unchecked, outpacing the immune system.

So far, it has been unclear whether the interactions between cancer and immune cells leave detectable genetic traces in the DNA of cells. By analysing the DNA of advanced cancers, researchers may be able to reconstruct how tumours interacted with the immune system in the past. These insights could reveal patterns that help predict a tumour's future behaviour and highlight new treatment opportunities.

Wang, Morison and Huang built a computer model to study how cancer and the immune system influence each other over time. By examining mutation patterns in tumours, they were able to trace how strongly the immune system shaped cancer growth.

The results showed that when cancer cells develop certain mutations that make them more visible to the immune system, the immune system launches strong bursts of activity to eliminate them. In this way, the immune system can shape which mutations persist in a tumour. For example, cancer cells with many mutations are more likely to be detected and destroyed. Over time, these strongly targeted cancer cells – and the immune cells that attack them – tend to disappear from the population. However, in some cancers, this effect is much weaker, and the immune system has less influence over the tumour's genetic changes.

A key challenge is understanding how cancer mutations arise in the first place. To make progress, researchers will need to combine large-scale population data with detailed single-cell data. This knowledge will be especially important for advancing immunotherapies and other precision cancer treatments.

---

Dunn et al. described the battle between the immune system and emergent tumours in three stages, termed the three Es of cancer immunoediting (*Dunn et al., 2004*). The first, Elimination, formulated by Burnet as the immunosurveillance hypothesis, states that the immune system can win this battle and eliminate small cancers (*Burnet, 1957*; *Burnet, 1967*). According to the 'bad luck' hypothesis (*Tomasetti and Vogelstein, 2015*), this is a frequent occurrence: only stochastically does the immune system allow cancers to sneak through (*George and Levine, 2020*). Should the cancer do so, it enters Equilibrium. Indeed, some cancers take years to grow to detectable size (*Dunn et al., 2004*), and there is evidence for small persistent tumours coevolving with the immune system (*Koebel et al., 2007*). Finally, Escape, when the cancer evolves mechanisms of evasion and grows to a size that can be detected (*Lakatos et al., 2024*). Deciphering genetic footprints from this coevolutionary process is of utmost clinical relevance from prognosis to treatment (*Lakatos et al., 2020*), especially since current measures such as immune infiltration, evidence of immune escape and tumour mutational burden are not foolproof markers of immunotherapeutic success and overall survival (*Zapata et al., 2023*).

Mutations accumulated during cancer evolution increase intratumour heterogeneity, providing a wide landscape of genotypes to improve the tumour's persistence (*Alexandrov et al., 2013*). Propensities for mutating more rapidly, increasing growth rate, developing metastases, and acquiring resistance to eventual treatment are possible consequences of this accrual of mutations (*Hanahan, 2022*). However, mutations may also alert the immune system of the presence of tumour cells and initiate

a suppressive response. This occurs when a mutation acquired by a cancer cell (called an antigenic mutation) results in the presentation by human leukocyte antigen (HLA) of immunogenic peptides called neoantigens at the cell surface (*McGranahan et al., 2017*). These neoantigens are recognisable by cytotoxic T lymphocytes (CTLs), specialised effector cells of the adaptive immune system, which bind to the presented neoantigen via a T cell receptor at their surface and kill the targeted cancer cell (*Schumacher and Schreiber, 2015*). While antigenic mutations are neutral in the absence of an immune response (*Wilkie and Hahnfeldt, 2013*), their fitness in general depends on the likelihood of neoantigen presentation and recognition by effector cells (*Łuksza et al., 2017*).

Antigenic mutations so targeted by the immune system are thus under negative selective pressure (*Van den Eynden et al., 2019*). The killing of cancer cells carrying antigenic mutations releases further neoantigens, leading to a positive feedback loop of generating immunity to cancer (*Chen and Mellman, 2013*). Cancer cells, however, may in turn combat the immune response via several mechanisms. Cancer cells can inhibit effector cells, such as by expression of programmed cell death-ligand 1, which normally presents in healthy cells to stop the attack of immune cells and can exhaust CTLs interacting with cancer cells that carry it (*Vinay et al., 2015*; *Leschiera, 2022*). Some cancer cells may escape the immune response, such as by reducing neoantigen presentation through down-regulating HLA (*McGranahan et al., 2017*) or by immune editing and losing antigenic mutations due to the immune negative selection (*Puleo and Polyak, 2022*). Cancer cells can even develop immune exclusion, physically restricting the immune cells' access to the tumour (*Hanahan, 2022*). These processes may arise individually or in concert, and they mean the immune system plays a crucial role in shaping a tumour's evolution (*Hanahan, 2022*; *Lakatos et al., 2024*).

Immunotherapies leverage the immune system to reverse the evolution of evasion by mitigating or interfering with these processes (*Leschiera, 2022*). Cytokine-based and tumour-infiltrating leukocyte-based immunotherapies help increase the effector cell population size (*Sotolongo-Costa et al., 2003*; *Schumacher and Schreiber, 2015*); immune checkpoint blockade therapies restore the effector response to immune-escaped cancer cells (*Chen and Mellman, 2013*; *Zapata et al., 2023*); other immunotherapies simply boost the ability of CTLs to kill tumour cells (*Schumacher and Schreiber, 2015*; *Heirene et al., 2025*). Understanding of the coevolution between cancer and immune cells, along with the cell-to-cell interactions that drive it, may inform when therapies will succeed—and why.

There has been a long history of studying antagonistic coevolution experimentally and theoretically (*Mode, 1958*; *Paterson et al., 2010*; *Yamamichi and Ellner, 2016*; *Huang et al., 2017*), with extensive literature on mathematical models of tumour–immune interactions (*Eftimie et al., 2011*; *Leschiera, 2022*; *Hamilton et al., 2022*). Deterministic models often describe the antagonistic relationship between cancer and immune cells as obeying Lotka-Volterra dynamics, with the immune system predating on its tumour prey (*d'Onofrio, 2005*; *Bozic et al., 2024*). Stochastic modelling of the tumour–immune system has been explored by e.g. George and Levine, who characterised escape as a random process with sequential mutations (*George and Levine, 2018*; *George and Levine, 2020*) and subsequently framed cancer evolution as an active optimisation process in response to an evolving immune landscape (*George and Levine, 2021*; *George and Levine, 2023*). In concert with patient sequencing data, Lakatos and colleagues proposed and applied a model of random antigenic mutation accumulation in order to describe the negative selection that neoantigens undergo, resulting in neutral-like evolutionary dynamics (*Lakatos et al., 2024*; *Lakatos et al., 2020*). Recently, Chen et al. included negative frequency dependence in a similar model of antigenic mutation accumulation, so that tumours are only immunogenic when a sufficiently large proportion of their cells present neoantigens, predicting that tumours undergoing this frequency-dependent selection have poorer treatment outcomes than their negative selection counterparts (*Chen et al., 2024*).

Coevolution between the immune system and the threats it faces has often been studied in a gene-for-gene framework, which centres on the genetic makeup of multiple populations being tracked (*Thompson and Burdon, 1992*). However, there is a dearth of stochastic models that explore the explicit evolutionary dynamics of both tumour and immune cell populations (*George and Levine, 2021*): the aforementioned models either encapsulate the immune response into a selection parameter $s$, implicitly assuming that effector cells react perfectly and instantaneously to a cancerous threat (*Lakatos et al., 2020*; *Chen et al., 2024*); or, they are deterministic and thus miss out the critical impact of random processes on small population sizes, while omitting direct genetic information with which sequencing data can be compared (*Korobeinikov et al., 2017*).

We address this gap in the literature by presenting and analysing a novel stochastic coevolutionary model of tumour–immune dynamics. As in Lakatos et al.'s model, antigenic mutations accrue in cancer cells undergoing a branching process and are negatively selected against by the immune system (*Lakatos et al., 2020*). The adaptive immune system is represented by specialised effector populations that react to emergent neoantigens (*Adam and Bellomo, 2012*), leading to complex dynamics on both tumour and immune fronts. We focus on interactions between effector cells and their targets, incorporating both active and passive recruitment of effector cells, killing of cancer cells, and inhibition of effectors by cancer cells, while including explicit mechanisms of escape. In particular, we inspect how these interactions are central in determining the evolution of the system. Stochastic simulations allow us to characterise the various dynamics and outcomes that emerge, along with informing the impact of immunotherapy.

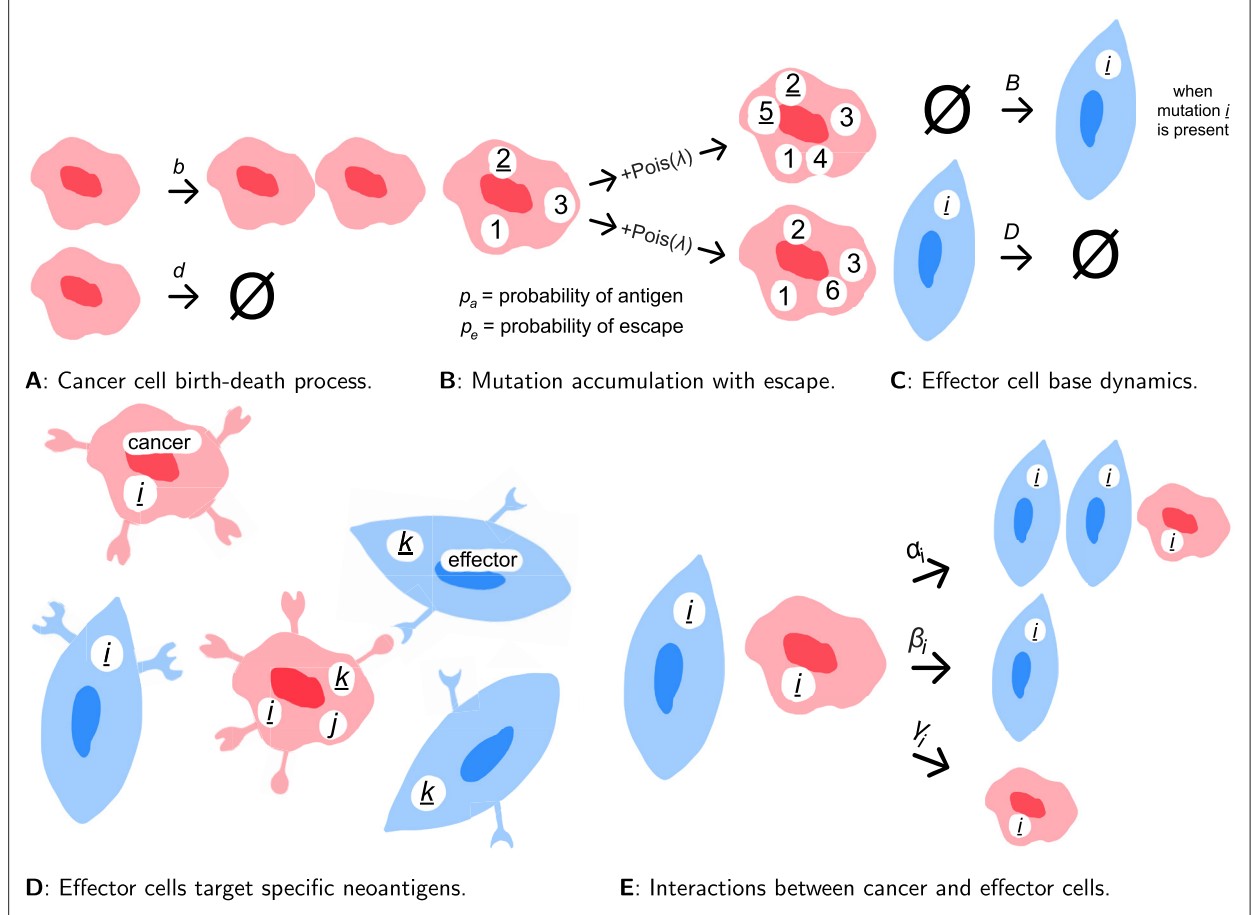

**Figure 1.** Cancer–immune stochastic model. (**A**) Cancer cells (red) stochastically divide and die with rates $b$ and $d$, respectively (here ∅ represents no cell). (**B**) During a division event, daughter cells inherit all of their mother cell's antigenic and neutral mutations, depicted by numbers (where underlined numbers are antigenic). Cells carrying antigenic mutations have a probability $p_e$ to escape and become neutral, as shown in the lower daughter cell with mutation 2. Each daughter cell also acquires a random number (drawn from a Poisson distribution with mean $\lambda$) of new mutations, where each mutation is antigenic with probability $p_a$ and neutral with probability $1 - p_a$. (**C**) For each antigenic mutation $i$ present in the system, a corresponding effector cell population $E_i$ exists (blue), whose size grows with constant rate $B$ and shrinks with per-capita death rate $D$. (**D**) Antigenic mutations in cancer cells (such as $i$ and $k$) display unique neoantigens at the cell surface, whereas neutral mutations (such as $j$) do not. The neoantigens can be identified by specialised effector cells, which can only interact with the corresponding cancer cells. (**E**) When a cancer cell carrying the antigenic mutation $i$ meets an effector cell of type $i$, three outcomes are possible: active recruitment of another effector cell of type $i$ with rate $\alpha_i$, killing of the cancer cell with rate $\beta_i$ and inhibition/exhaustion of the effector cell with rate $\gamma_i$. The expressions for the rates are found in *Equation 1*.

## Results

### Modelling specialised effector response and cancer–immune interactions

Consider a population of cancer cells undergoing a stochastic branching process with per-capita birth rate $b$ and per-capita death rate $d$ (see *Figure 1A*). We assume births and deaths happen randomly based on their rates, with exponentially distributed wait times between events, as in the Gillespie algorithm (*Gillespie, 1976*). At each cancer cell division, daughter cells inherit their mother cell's mutations and acquire a random number of new mutations, drawn from a Poisson distribution with mean $\lambda$ (*Lederberg, 1989*), as shown in *Figure 1B*. Similar to Lakatos et al., we focus on the exomic region (*Lakatos et al., 2020*), where the value of $\lambda$ ranges between 1 and 10. All mutations are unique, as in the infinite sites approximation, where the exome is considered long enough for two co-occurring point mutations to be negligible (*Kimura, 1969*). We distinguish between two types of mutations: with probability $p_a$, a mutation is antigenic and can be recognised by given effector cells, and with probability $1 - p_a$ it is neutral (*Schumacher and Schreiber, 2015*; *Lakatos et al., 2020*). Here, we write the antigenic mutations of a cancer cell (labelled by $\ell$) as $M_{a,\ell}$ and the neutral mutations as $M_{n,\ell}$, where $M_\ell = M_{a,\ell} \cup M_{n,\ell}$ is the set of all mutations carried by a cancer cell. During a division, there is a probability $p_e$ that an antigenic mutation escapes the immune system, therefore making the cell possessing it (and all of its descendants *Baar et al., 2016*; *Chen et al., 2024*) undetectable to the immune system (see *Figure 1B*), rendering their antigenic mutation sets empty. It is important to note that immune escape is not necessarily a permanent state; cells that have undergone escape may subsequently acquire new antigenic mutations in future divisions, thereby regaining susceptibility to immune detection.

Neoantigens trigger responses from the adaptive immune system: each antigenic mutation $i$ calls a unique, specialised effector population, as in gene-for-gene coevolution (*Thompson and Burdon, 1992*). We will write $(E_i)$ to denote an effector cell of type $i$ and $E_i$ to represent the corresponding population size (and, in an abuse of notation, occasionally the population itself, when this will not cause too much confusion). Whenever mutation $i$ exists within the population, effector cells of type $i$ are passively recruited from the body at constant rate $B$ ($\varnothing \xrightarrow{B} (E_i)$, for $\varnothing$ the absence of a cell, with rate $B$), and die with per-capita rate $D$ (*Kuznetsov et al., 1994*; *Morselli et al., 2024*), as shown in *Figure 1C*.

To each antigenic mutation $i$, we associate two random numbers, each drawn independently from an exponential distribution with mean 1: an antigenicity $A_i$, describing the propensity for an effector cell $(E_i)$ to kill a cancer cell possessing mutation $i$, and an immunogenicity $I_i$, which relates to the rate at which effector cells of type $i$ are recruited during the specified tumour–immune interaction (see equation (*Equation 1*) and *Figure 1E*). These can be thought to encapsulate the probability of an antigenic mutation leading to the presentation of neoantigens by HLA molecules and subsequent recognition by CTLs (*Łuksza et al., 2017*).

Correspondingly, when a cancer cell with an antigenic mutation presents a neoantigen at its surface, only effector cells of the corresponding population can interact with it (*Leschiera, 2022*), as shown in *Figure 1D*. We will write $(C_i)$ to denote a cancer cell possessing an antigenic mutation $i$ and $C_i$ for the corresponding population size. Interactions between cancer cells $(C_i)$ and effector cells $(E_i)$ have three possible outcomes: active recruitment of another effector cell $(E_i)$ with rate $\alpha_i$, killing of the cancer cell $(C_i)$ with rate $\beta_i$ and inhibition/exhaustion of the effector cell $(E_i)$ with rate $\gamma_i$ (see *Figure 1E*). The precise description of the system can be described by the following set of microscopic reactions:

$$
\begin{aligned}
(C_{M_\ell}) &\xrightarrow{b} (C_{M_\ell \cup M'}), (C_{M_\ell \cup M''}) \\
(C_{M_\ell}) &\xrightarrow{d} \emptyset \\
\emptyset &\xrightarrow{B} (E_i) \\
(E_i) &\xrightarrow{D} \emptyset \\
(C_i), (E_i) &\xrightarrow{\alpha_i} (C_i), (E_i), (E_i) \\
(C_i), (E_i) &\xrightarrow{\beta_i} (E_i) \\
(C_i), (E_i) &\xrightarrow{\gamma_i} (C_i)
\end{aligned}
\tag{1}
$$

where $\alpha_i = \frac{\alpha_0 I_i}{1+\alpha_0 h_\alpha C_i I_i}$, $\beta_i = \beta_0 A_i$, $\gamma_i = \gamma_0$, and $\alpha_0$, $\beta_0$, $\gamma_0$, $h_\alpha$ are constants for the entire tumour. For simplicity, we will call $\alpha_0$, $\beta_0$ and $\gamma_0$ the recruitment, killing and inhibition rates, respectively, even though they are not strictly rates. We have denoted the cell carrying a set of mutations $M_\ell$, which can be partitioned into antigenic mutations $M_{a,\ell}$ and neutral mutations $M_{n,\ell}$, by $(C_{M_\ell})$. Upon division, each daughter cell inherits the full set of mutations from the mother cell and independently acquires additional new mutations, denoted by $M'$ and $M''$ for the two daughter cells, respectively. The numbers of newly acquired antigenic and neutral mutations for each daughter cell are drawn independently from Poisson distributions with means $p_a\lambda$ and $(1-p_a)\lambda$, respectively. With probability $1-p_e$, the daughter cell retains the antigenic and neutral mutations of the mother cell. With probability $p_e$, the daughter cell undergoes immune escape, and all mutations (including those inherited and newly acquired) are considered neutral, resulting in empty antigenic mutation set. Note that the reactions of active recruitment and inhibition/exhaustion have opposite effects on the effector cell population; here, we assume $\alpha_0 > \gamma_0$, the opposite of which is rarely considered (*Nani and Freedman, 2000*), as it leads to net decreasing effector population sizes upon interactions with cancer cells. However, active recruitment $\alpha_i$ obeys a type-II functional response rather than a type-I (i.e. linear) response as inhibition/exhaustion $\gamma_i$; this is because there is an upper bound to how quickly new effector cells can be recruited (*Holling, 1959*; *Kuznetsov and Knott, 2001*).

We are interested in genetic information relating to each of the two populations, which we will use to identify and measure the coevolution between effector and cancer cells. For the cancer cells, we define the following summary statistics: the site frequency $S_j$, denoting the number of antigenic mutations present in $j$ cells, and the single-cell mutational burden $B_k$, the number of cells that possess $k$ antigenic mutations. The conglomeration of these (for non-negative integers $j$ and $k$, respectively) forms the site frequency spectrum (SFS) and the single-cell mutational burden distribution (MBD). These satisfy $\sum_{j=1}^{C} jS_j = \sum_{k=1}^{M_a} kB_k$, where $C$ denotes the total population of cancer cells and $M_a$ denotes the total number of antigenic mutations across all cancer cells (*Morison et al., 2023*). We will write $U$ for this quantity (i.e., for either side of the previous equality), the total number of antigenic mutational occurrences.

For the effector population, we define the average antigenicity $\langle A \rangle$ and the average immunogenicity $\langle I \rangle$ as follows:

$$\langle A \rangle = \frac{\sum_{i=1}^{M_a} A_i E_i}{\sum_{i=1}^{M_a} E_i} \quad \text{and} \quad \langle I \rangle = \frac{\sum_{i=1}^{M_a} I_i E_i}{\sum_{i=1}^{M_a} E_i}. \tag{2}$$

Models that focus only on the cancer cell populations ascribe an antigenicity to the tumour itself (*Lakatos et al., 2020*; *Chen et al., 2024*); here, by considering both cancer and effector populations, we can also average the antigenicities and immunogenicities over all effector cells. By choosing to average over the effector cells as in *Equation 2*, $\langle A \rangle$ and $\langle I \rangle$ become measures of the gene-for-gene immune response and its potency in fighting its tumour target, providing an additional angle on the coevolutionary system not present in single-population models.

## Cyclic dynamics between antigenicity and immunogenicity

Antagonism between effector cells and their cancer targets results in a range of complex dynamics. For elevated immune efficacy such as high recruitment and killing rates $\alpha_0$ and $\beta_0$, emerging tumours are suppressed in early stages. However, when the probability of escape $p_e$ is large enough, then neutral-like dynamics ensue, as most cancer cells evade the immune response. While it is expected that negative selection of the immune system on new antigens will selectively prune cancer cells with more antigenic mutations (*Lakatos et al., 2020*), this process depends on the effector populations themselves, through the interactions formulated in equation (*Equation 1*). Since here we explicitly model both tumour and immune cell types, we are able to explore their population dynamics in concert.

Indeed, we can clearly observe a dynamical response between the cancer and effector cells in single evolutionary trajectories. *Figure 2* depicts two representative realisations of our stochastic simulations, one for a lower mutation rate ($\lambda = 1$) and one for a high mutation rate ($\lambda = 10$), the latter of which can be thought of as representing hyper-mutated tumours (*Lakatos et al., 2020*). The population dynamics of *Figure 2A, C* show the effector population undergoing irregular spikes, following

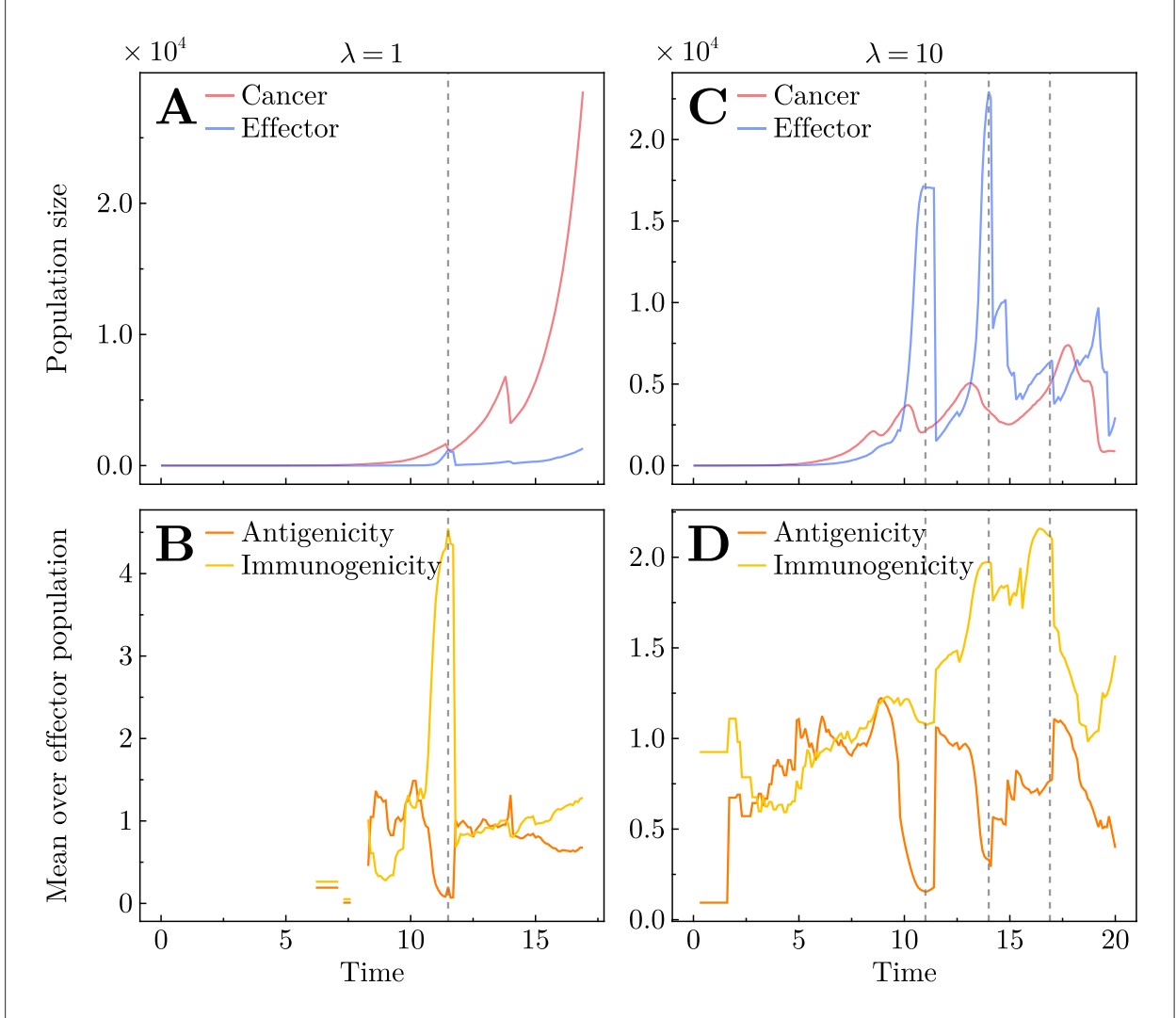

**Figure 2.** Single stochastic realisations for mutation rate $\lambda = 1$ (**A, B**) and $\lambda = 10$ (**C, D**). (**A, C**) Population dynamics, where red and blue lines depict total cancer and effector cell populations. (**B, D**) Average immunogenicity (yellow line) and antigenicity (orange line) in the effector population for the corresponding realisation. Vertical dashed grey lines in all panels indicate the timing of effector population spikes, which are consistent with the vertical dashed lines in the Muller plots (*Appendix 1—figure 3*) showing the dynamics of individual effector phenotypes over time for the same realisations. Interaction parameters: recruitment rate $\alpha_0 = 0.03$ and killing rate $\beta_0 = 0.3$ (**A, B**) and $\alpha_0 = 0.005$ and $\beta_0 = 0.01$ (**C, D**).

classic in-phase cycles with the cancer population as seen in other antagonistic systems (*Abrams, 2000*).

Higher mutation rates lead to the appearance of more mutants, thus potentially more new types of cancer–immune interactions. Based on previous predator–prey studies (*Yoshida et al., 2003*; *Yamamichi, 2020*), we developed a method to quantify the number of cancer–immune cycles at low and high mutation rates. Different from typical predator–prey systems where the two antagonistic species fluctuate between relatively stable ranges of population sizes (*Yoshida et al., 2003*), the abundances of effector and cancer cells in our system often have increasing trends, as seen in *Figure 2A, C*. Consequently, the phase portraits of the abundances of cancer effectors show stochastic and irregular behaviour (see *Appendix 1—figure 4*), rather than having a closed or open oval shape as in predator–prey systems. This makes it hard to quantify whether a cycle is in-phase or out-of-phase using the shape of the phase portraits as in predator–prey systems (*Jones and Ellner, 2007*; *Papkou et al., 2016*), although visually out-of-phase cycles are rare in our simulations. Instead, we develop a method to quantify the number of cancer–immune cycles in our simulations by tracking the directional changes in phase portraits, validated by using a non-evolving stochastic predator–prey system as a control (see

*Appendix 1—figure 5*). As expected, the number of cycles increases when the mutation rate is higher (see *Appendix 1—figure 6*). The majority are counter-clockwise cycles, where the cancer population increases first, followed by the increase of the effector population (see *Appendix 1—figure 7*). However, we also observe a small fraction of clockwise cycles, especially when mutation rate is higher. Clockwise cycles have been observed in various predator–prey systems and could arise as a consequence of coevolution (*Cortez and Weitz, 2014*).

Because of the specialised nature of our model, these cyclic dynamics arise when a single antigenic mutation $i$ causes the rapid active recruitment of the corresponding effector cells ($E_i$) to combat the subclone possessing mutation $i$ during that time period. Once mutation $i$ is eradicated from the cancer cell population, the effector cells ($E_i$) specialised to their neoantigen are removed from the system, as they no longer play a dynamical role and die out exponentially (see *Figure 1C*). The expected frequency and amplitude of these effector spikes can be approximated, as described in section Stochastic analysis of Appendix. We validate this by inspecting the coevolution of antigenicity and immunogenicity in the corresponding single realisations. In *Figure 2*, we can see that the effector spikes arise due to the emergence of one or several mutations that have a large immunogenicity $I_i$ (*Appendix 1—figure 3*) dominating the average immunogenicity in the effector population (*Figure 2B, D*). Unexpectedly, in addition to this elevated immunogenicity, *Figure 2B, D* as well as *Appendix 1—figure 3* imply that these spikes also arise for mutations with low antigenicity. These fluctuations in $\langle I \rangle$ and $\langle A \rangle$ are coupled to the spikes in the population size (*Figure 2* and *Appendix 1—figure 3*); had a mutation emerged with a large antigenicity, it would have been quickly eradicated, and so the effector population size would not have grown sufficiently to be identified as a spike. As expected, the higher mutation rate ($\lambda = 10$) leads to more cycles of spikes not only in the cancer–effector dynamics but also in the coevolution of antigenicity and immunogenicity dynamics.

## Interactions dictate the outcome of tumour progress

The outcome of the coevolution between the immune system and a tumour depends strongly on the interaction parameters between effector and cancer cells (*Equation 1*). *Figure 3A, D* illustrates heat maps for the tumour suppression proportion across different values of the active effector recruitment rate $\alpha_0$ and the cancer killing rate $\beta_0$. Due to the stochastic nature of the model, the outcome for a given parameter set is probabilistic. As expected, for higher values of $\alpha_0$ and $\beta_0$, more effector cells are recruited and more cancer cells are killed, thus the suppression increased. While this pattern holds for different mutation rates (*Figure 3*), it is more dominating when mutation rate is high. For a higher mutation rate ($\lambda = 10$), the immune system is more effective since there are more antigenic mutations to target, resulting in a high suppression (see *Figure 3D*). For a lower mutation rate ($\lambda = 1$), however, only a small proportion of tumours are suppressed even with strong immune response (*Figure 3A*). This is because only a fraction of the population is antigenic (and thus undergoing immunoselection): observe *Appendix 1—figure 10A–C versus D–F*. It is also worth noting that in this case most suppressed tumours go extinct at early times (*Appendix 1—figure 9A*) due to stochasticity.

When the active effector recruitment rate is smaller than the effector inhibiting rate $\alpha_0 < \gamma_0$ (*Figure 3*, red dashed line), the net outcome for effectors during an interaction is negative. This implies that the resulting suppressed tumours in this parameter region were either defeated by a passively recruited effector population, or by effector types $i$ corresponding to antigenic mutations with particularly high immunogenicities $I_i$. This is possible since $I_i$ is drawn from an exponential distribution (and thus can be high), meaning that even when $\alpha_0$ is low, the active recruitment can be greater than the inhibition/exhaustion by chance. This is exactly what we observed in our stochastic simulations. Under high mutation rate ($\lambda = 10$), most tumours are not suppressed under low recruitment and killing rates $\alpha_0$ and $\beta_0$ (*Figure 3D*), whereas for high interaction parameters most tumours go extinct (*Figure 3E*) rather than maintaining a slow growing pace (*Figure 3F*). Again, we see a similar pattern under a low mutation rate ($\lambda = 1$), though more weakly and with more noise.

Interestingly, for a high mutation rate, there exists an intermediate range of interaction parameters that allows for tumours to neither go extinct nor to grow to capacity (*Figure 3F*). One further interpretation of *Figure 3D* is the presence of a killing threshold (here, at $\beta_0 \approx 10^{-1}$) above which tumours are suppressed, no matter the active recruitment rate $\alpha_0$. The rest of the domain of *Figure 3D* then exhibits a much stronger dependence on the active recruitment rate $\alpha_0$, in line with the results of *Wilkie and Hahnfeldt, 2013*.

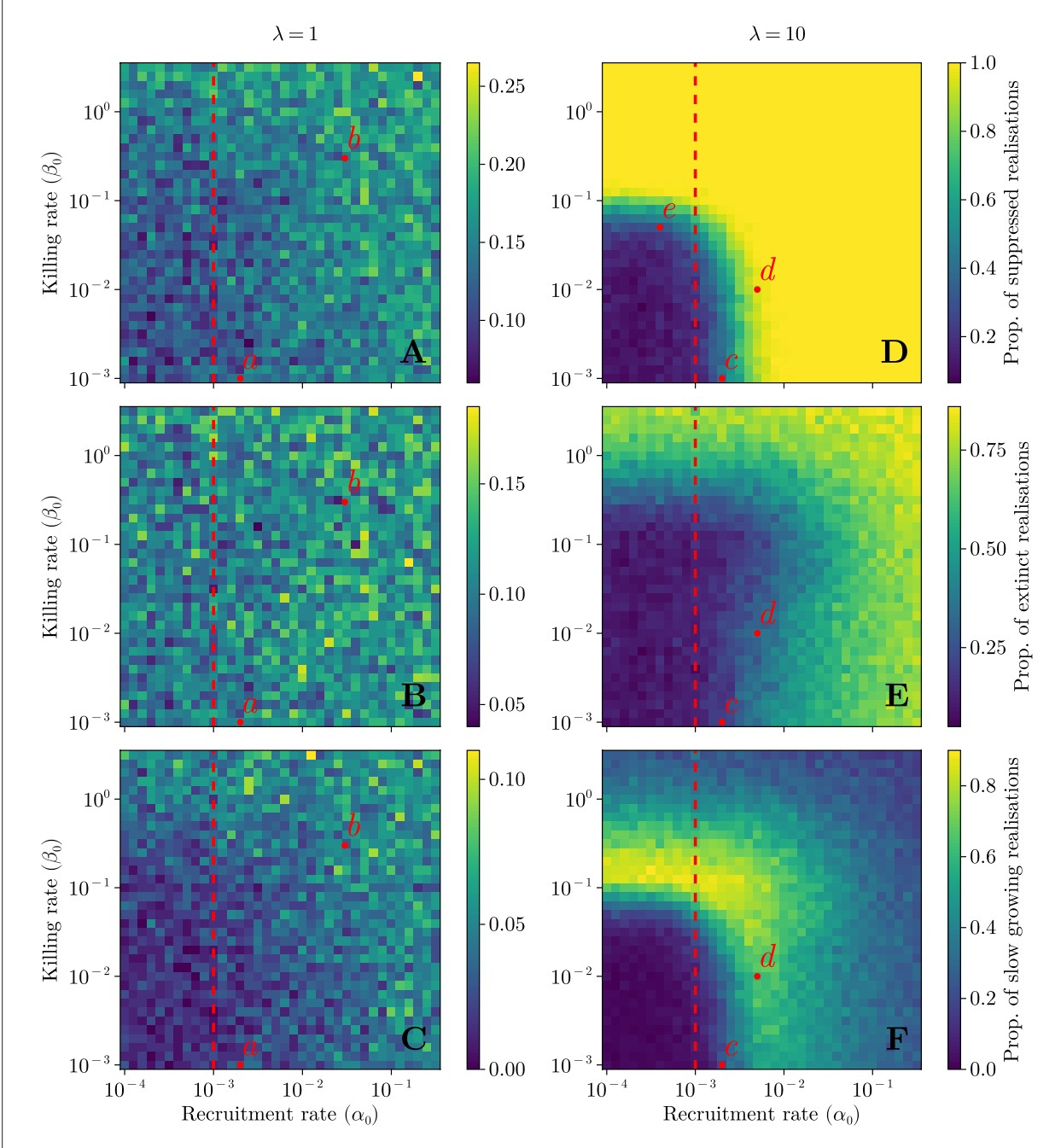

**Figure 3.** Outcome heat maps for mutation rate $\lambda = 1$ (**A–C**) and $\lambda = 10$ (**D–F**) for tumours interacting with an immune system characterised by cancer cell–effector cell interaction parameters $\alpha_0$ (active recruitment of effectors) and $\beta_0$ (killing of cancer cells). (**A, D**) Proportion of suppressed tumours. (**B, E**) Proportion of extinct tumours. (**C, F**) Proportion of slow-growing tumours: tumours that are suppressed but do not go extinct. Points *a* and *b* (*c* and *d* for $\lambda = 10$) label parameter sets of low and high immune effectiveness, respectively. The red dashed line denotes inhibition rate $\gamma_0$ (here $\gamma_0 = 10^{-3}$); see the Discussion section for details of point *e*. All parameter values not specified here are listed in **Table 1**.

## Genetic markers of selection

Stochastic mutation accumulation in an exponentially growing population has been widely studied (**Durrett, 2013**; **Bozic et al., 2016**). When the immune system has little impact on the cancer cell population, therefore, it is unsurprising to see consistent increases in the average number of mutations per cancer cell, as in **Figure 4A**. The theoretical expectation of neutrally accumulated mutations can be approximated by the average flux of number of new mutations entering the system, $2bt$

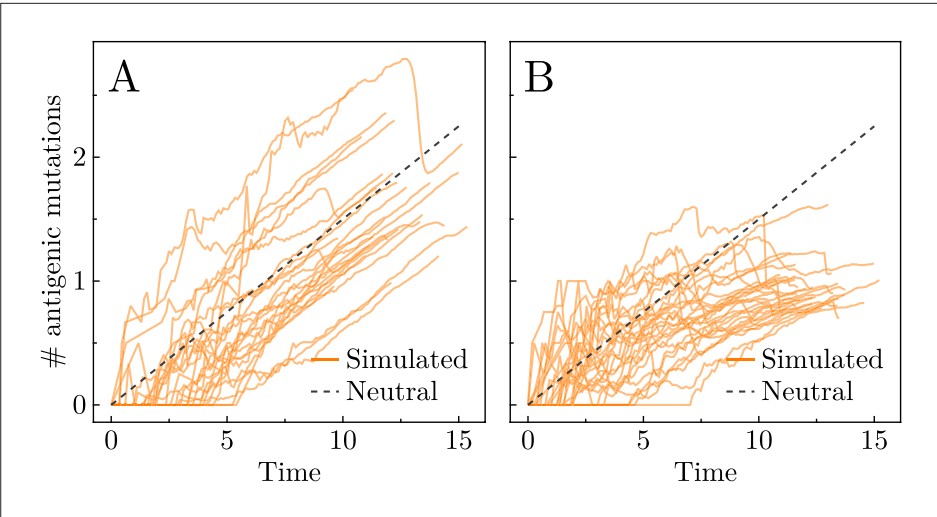

**Figure 4.** Average number of antigenic mutations per cell (solid orange lines) for several representative realisations when $\lambda = 1$. Theoretical prediction for the accumulation of neutral mutations per cell in an exponentially growing population is shown in grey dashed line. (**A**) Low immune effect: recruitment rate $\alpha_0 = 0.002$ and killing rate $\beta_0 = 0.001$. (**B**) High immune effect: $\alpha_0 = 0.03$ and $\beta_0 = 0.3$. (Parameter sets chosen as points $a$ and $b$ from **Figure 3A**.)

(**Samuels, 1971**), as plotted in **Figure 4**. (Note that a correction for large times has recently been shown to apply **Stein and Werner, 2025**; **Cheek and Johnston, 2023**, which helps explain why the dashed line over-estimates the simulated data in **Figure 4**.) When the effector population is selectively killing, however, possessing more antigenic mutations makes a cancer cell more likely to be killed, so the average number of antigenic mutations per cancer cell does not continuously increase with the population, as shown in **Figure 4B**. This effect of selection is even more pronounced in the case of a higher mutation rate ($\lambda = 10$), as shown in **Appendix 1—figure 12**.

The average number of antigenic mutations per cell is the mean of the MBD: $\langle B \rangle = U/C$, which, like the SFS, can be extracted from bulk data. However, single-cell sequencing data can also provide information on the MBD overall, which can be used in combination with bulk information to infer what selection is taking place in the system (**Moeller et al., 2024**). As discussed in section Stochastic analysis of the Appendix, the expected neutral distributions of the SFS and the MBD have been solved (**Gunnarsson et al., 2021**; **Morison et al., 2023**), so divergence from these may demonstrate the presence of selection and the strength of cancer–immune interactions. In **Figure 5**, the SFS and MBD averaged over 100 realisations are plotted in conjunction with the corresponding theoretical predictions (black dashed line) for the case with no immune response (that is, where all mutations are neutral). The first and third rows (green points) represent neutral mutations, whereas the second and fourth rows (orange points) measure antigenic mutations; for the MBDs, the mean (that is, the value $U/C$) was plotted in dashed vertical lines for both the simulated data (in green and orange) and the theoretical predictions (in grey), which were calculated with **Equations 3 and 4** of the Appendix.

We notice that in the case of low immune effectiveness (**Figure 3A–D**), there is little deviation from the neutral expectation. When the immune system plays a larger role, however, the distinction is significant, as in **Figure 3E–H**. In particular, the cells with more antigenic mutations were more selectively killed by the immune system, so that the tail of the MBD is smaller and the mean is shifted down, as seen most prominently in **Figure 3H**. As before, when the mutation rate is higher ($\lambda = 10$), these effects are more striking, as shown in **Appendix 1—figure 13**. On the other hand, the SFS shows limited difference from its neutral expectation (**Figure 3A, E**), reiterating the importance of integrating single-cell data with bulk sequencing data in identifying immune effects. Under stronger negative selection, we expect a depletion of antigenic mutations from the population and thus lower site frequencies than the theoretical prediction (**Lakatos et al., 2020**). This is visible as a slight depression of the data compared to the neutral prediction in **Figure 3F**, though such an observation is much clearer for a faster immune response, as discussed in section Stochastic analysis.

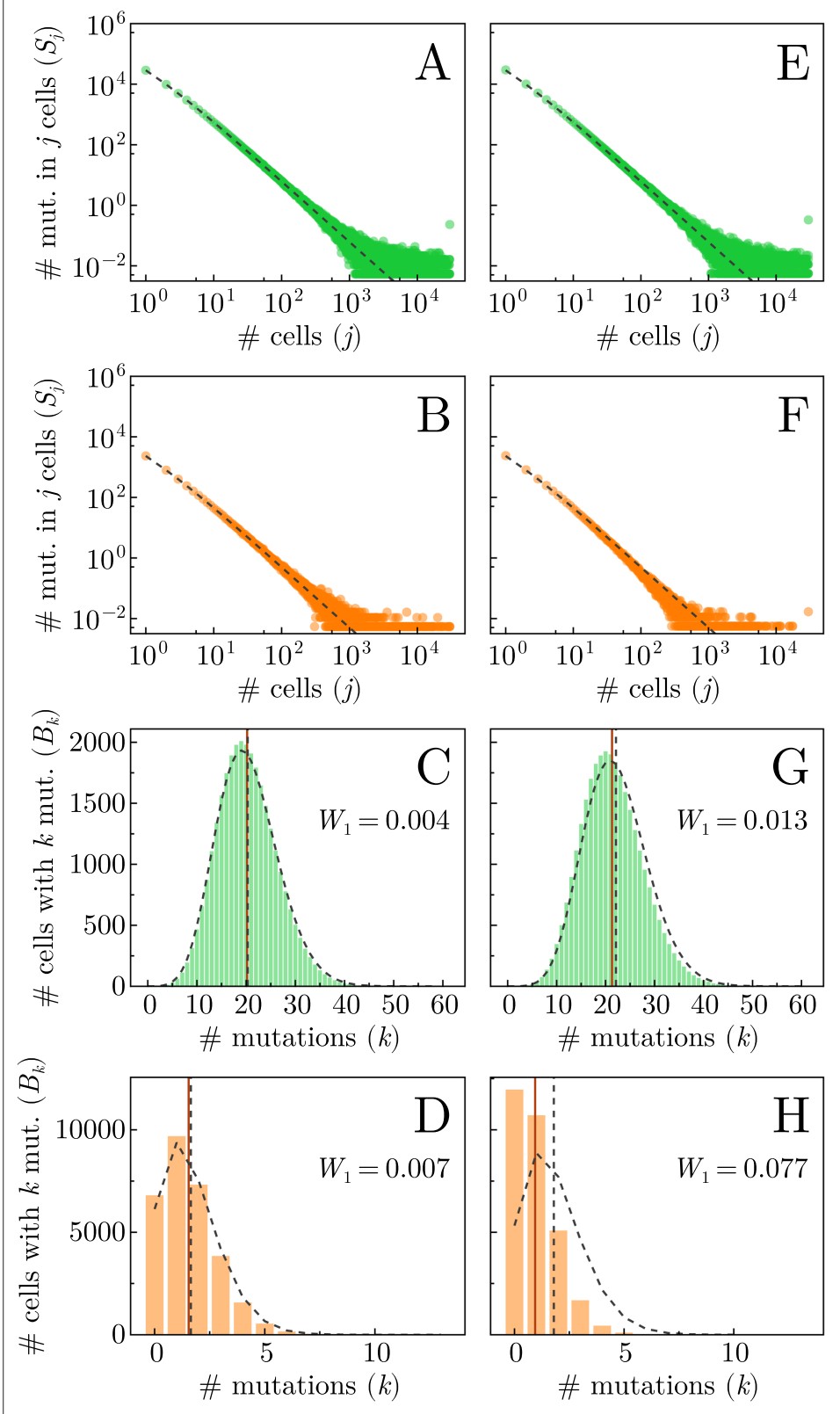

**Figure 5.** Genetic markers of selection and cancer–immune interactions for $\lambda = 1$. Panels show the site frequency spectrum (SFS) for neutral (**A, E**) and antigenic mutations (**B, F**), and the single-cell mutational burden distribution (MBD) for neutral (**C, G**) and antigenic mutations (**D, H**). Left-column panels correspond to low immune effectiveness ($\alpha_0 = 0.002$, $\beta_0 = 0.001$, point a in **Figure 3**), while right-column panels correspond to high immune

*Figure 5 continued on next page*

*Figure 5 continued*

effectiveness (recruitment rate $\alpha_0 = 0.03$, killing rate $\beta_0 = 0.3$, point b in **Figure 3**). Black dashed lines indicate theoretical predictions under neutral evolution (no immune response). Solid and dashed vertical lines in panels **C**, **D**, **G**, and **H** denote the means of the simulated and theoretical MBDs, respectively. The $W_1$ value represents the Wasserstein distance between the simulated and theoretical MBDs with the number of mutations $k$ is rescaled to the interval [0,1], and the number of cells $B_k$ is normalised to form a probability density. Results are averaged over 100 realisations; all other parameter values are given in **Table 1**.

## Robustness of our observations

We have shown systematically how the two key parameters, i.e. the active effector recruitment rates $\alpha_0$ and the cancer killing rate $\beta_0$, impact the early suppression of tumour cells (**Appendix 1—figure 9**), different outcomes in tumour progression (**Figure 3**), as well as the proportion of antigenic cells in tumours (**Figure 3**). While there is high stochasticity in the system, we observe consistent patterns in evolutionary outcomes across the parameter space. We then pick up representative parameters from distinct parameter regions (e.g. points $a$ and $b$ in **Figure 3**) to compare the averaged distributions of genetic markers—SFS and single-cell MBDs—between these regions based on many realisations (**Figure 3**). Using neutral expectations without immune–cancer interactions as the baseline, we see larger differences (measured by the Wasserstein distance) in single-cell mutation burden distributions (**Figure 3** and **Appendix 1—figure 13**, C vs. G, D vs. H) in high immune effectiveness (point $b$, **Figure 3**) compared to lower immune effectiveness (point $a$, **Figure 3**). We further confirm the robustness of our observations in the whole parameter space using the values of Wasserstein distance. The parameter regions with smaller or larger differences in the single-cell mutation burden distribution compared to the neutral expectation (**Appendix 1—figure 14**) are consistent with the patterns reflected by the tumour progression (**Figure 3**).

## Discussion

Coevolution between effector cells and their cancer targets sets the stage for the emergence and subsequent development of a tumour. Based on expansive literature in this field (**d'Onofrio, 2005**; **Lakatos et al., 2020**; **George and Levine, 2018**), we focus on important perspectives which have not yet been addressed by the previous models: in particular, the stochastic nature of these early-stage small cell population sizes as well as explicit interactions between cancer cells and immune cells. Here, we model cancer–immune coevolution, wherein specialised effector cells respond to the presence of randomly accumulated neoantigens in a growing cancer population. We uncover a variety of cancer–immune population dynamics, from the escape or extinction of the tumour to cycles characteristic of antagonistic interactions. We find that the suppression of the tumour by the immune system depends strongly on the cancer–immune interaction parameters, as well as rates of antigenic mutation accumulation in the cancer population. Using mutational distributions, we identify selection and the strength of cancer–immune interactions in the system, arguing for the importance of integrating population- and single-cell-level data, especially in the context of informing immunotherapeutic practices with model predictions.

Instead of encapsulating the immune impact into a selection parameter, which assumes the effector population reacts immediately and perfectly to any new antigenic mutation (**Lakatos et al., 2020**; **Chen et al., 2024**), we model the explicit interactions between cancer and immune cells. Our model unveils effector population dynamics during burst-like responses to growing subclones of the cancer population that possess antigenic mutations with high immunogenicity (see **Figure 2**). These immune population spikes serve to eliminate specific mutations from the cancer population. Via this selective killing of cancer cells, a process known as immunoselection (**Schumacher and Schreiber**, **2015**), the immune system moulds the genetic landscape of the tumour in ways that are identifiable via sequencing data. For instance, the decrease in average antigenic mutations per cell (see **Figure 4**) and the truncation of the high-burden tail of the MBD (see **Figure 5**) demonstrate that cells with more mutations face stronger negative selection by immune response and are thus pruned from the population. It should be noted, however, that for certain cancers the neoantigenic landscape has been found to be only weakly impacted by cancer–immune coevolution (**Lakatos et al., 2024**). Central to this modelling challenge is understanding the mutational process itself.

A fraction of somatic mutations arising in cancer populations gives rise to the presentation of neoantigens (*Linnemann et al., 2015*). This is a random process, wherein high mutational loads do not necessarily correspond to high antigenicities (*Schumacher and Schreiber, 2015*). This implies that careful consideration of the mutational burdens of cells as well as the antigenicities of individual mutations is crucial to understanding the resulting evolutionary dynamics of the system. The total mutational burden of the tumour, however, is not a sufficient predictor of response to treatment unless mutations that have escaped are taken into consideration (*Zapata et al., 2023*). While the SFS and the single-cell MBD can inform and help quantify selection (see *Figure 5* and *Appendix 1—figure 13*), more work needs to be done to understand the impact of different mechanisms of immune escape on genetic data.

Informing treatment is one of the principal tenets of mathematical modelling in oncology (*Gatenby and Vincent, 2003 Bozic et al., 2024*). Some immunotherapies increase the ability of effector cells to kill cancer cells (*Schumacher and Schreiber, 2015*), while others, termed immune checkpoint inhibitors, reactivate immune predation in the case of antigenic mutations having escaped detection (*Zapata et al., 2023*). The neoantigens accumulated after escape thus work against the cell once immunotherapy renders the cell visible to the immune system once more, though immune evasion may still impede immunotherapy (*George and Levine, 2018*). If, however, the antigenicities of the cancer cells are low due to immunoselection, the tumour will be less likely to respond well to immune checkpoint inhibition (*Zapata et al., 2023*).

Few models have explored the relative advantages of different changes in tumour–immune interactions, which represent the impact of immunotherapies discussed above (*Morselli et al., 2024*). Wilkie and Hahnfeldt, for instance, demonstrated that resistance to immune predation plays a smaller role than effector recruitment (*Wilkie and Hahnfeldt, 2013*). Our results show that these relative advantages are highly dependent on the system itself: in *Figure 3D*, moving upwards from point $c$ (decreasing the resistance to predation) has little impact on outcome, whereas moving to the right (increasing recruitment) changes the outcome drastically. Conversely, at point $e$, we notice the opposite effect: a change in predation resistance impacts the outcome but a change in recruitment does not. Importantly, only treatments that transform system parameters can succeed in a robust fashion, since only changing the state will still result in the same equilibria as before (*Gatenby and Vincent, 2003*).

Limitations exist when trying to model the cancer–immune system. When the (antigenic) mutation rate is low, the fraction of the tumour visible to the immune system is too (see *Appendix 1—figure 10*). The cancer population dynamics are therefore largely neutral (*Lakatos et al., 2020*), though our model reveals complex effector dynamics. One can also assume a certain antigenic proportion in the tumour before immune recognition (*Chen et al., 2024*), or address the growth-threshold conjecture, which states that the immune system will respond to a large enough tumour growth rate, rather than a certain tumour size (*George and Levine, 2018*; *Grossman and Paul, 1992*). The situation can also be further complicated by explicitly considering the composite state of an effector cell in the process of killing its cancer prey as a new conjugate type in the model, as has recently been done by Yang et al., who demonstrated its possible impact on the resulting dynamics, including on the outcome and its time scale (*Yang et al., 2025*).

By employing an individual-based model, we can compare expected mutational distributions with corresponding genetic data. The signatures of selection and strength of cancer–immune interactions in the system, along with a mechanistic knowledge of these interactions, may help inform us of a tumour's evolutionary history, along with its immunotherapeutic potential.

# Materials and methods
## Stochastic simulation

We implemented our model using a standard Gillespie algorithm (*Gillespie, 1976*). Each cancer cell is represented as an individual entity characterised by its unique spectrum of mutations. Similarly, each type of effector cell is modelled as a distinct entity that interacts specifically with cancer cells carrying the corresponding antigenic mutation. When a cancer cell divides, new antigenic mutations may arise in the daughter cells, potentially leading to the emergence of new effector cell types. Consequently,

**Table 1.** Notation and baseline parameter values.

| Symbol | Baseline | Description |
|---|---|---|
| $b$ , $d$ | 1 , 0.1 | Birth and death rates of a cancer cell |
| $\lambda$ | 1 , 10 | Mean number of exomic mutations acquired per division per daughter cell |
| $p_a$ | 0.075 | Probability of a mutation being antigenic (rather than neutral) |
| $p_e$ | $10^{-6}$ | Probability of escape of an antigenic cancer cell into neutrality |
| $C_i$ | | Population of cancer cells carrying antigenic mutation $i$ , with total cancer cell population size $C$ |
| $E_i$ | | Population of effector cells of type $i$ , with total effector cell population $E$ |
| $B$ , $D$ | 0.2 , 0.1 | Passive recruitment and per-capita death of an effector cell |
| $\alpha_i$ , $\beta_i$ , $\gamma_i$ | —, —, $10^{-3}$ | Active recruitment, killing and inhibition/exhaustion rates, respectively, for interactions between cancer and effector cells of type $i$ |
| $A_i$ , $I_i$ | | Antigenicity and immunogenicity, respectively, of antigenic mutation $i$ (each drawn from $\mathrm{Exp}(1)$), with averages over the effector population $\langle A \rangle$ and $\langle I \rangle$ |
| $S_j$ | | Site frequency: number of antigenic mutations occurring $j$ times |
| $B_k$ | | Single-cell mutational burden: number of cancer cells with $k$ antigenic mutations |
| $M_{a,\ell}, M_{n,\ell}$ | | Set of antigenic and neutral mutations, respectively, carried by a cancer cell $\ell$ |
| $M_a, M_n$ | | Total number of all antigenic and neutral mutations: $M_a = \lvert \bigcup_\ell M_{a,\ell} \rvert$ and $M_n = \lvert \bigcup_\ell M_{n,\ell} \rvert$ |
| $U$ | | Total antigenic mutational occurrences: $U = \sum_{j=1}^{C} jS_j = \sum_{k=1}^{M_a} kB_k$ |
| $K$ | $3 \times 10^4$ | Ending cancer population size for stochastic simulations |
| $T_{\mathrm{end}}$ | 16 | Maximum time for a simulation |

as the tumour population expands, both the number of cancer cells and the diversity of effector cell types increase.

In the standard Gillespie algorithm, the number of possible 'reactions' scales quadratically with the number of cancer cells due to the specialised interaction (defined in *Equation 1*), resulting in computational costs that become prohibitive for simulating large tumour populations. To address this challenge, we developed a dynamic dependency graph that tracks the dependencies between reactions and cell types, enabling efficient reuse of computations from previous steps. This optimisation reduces the computational cost per step to a constant, independent of the number of cancer cells.

A realisation ends when either the cancer cell population size exceeds a threshold $K$, which we call *no suppression*; goes *extinct*; or the cancer cell remains below $K$ at time $T_{\mathrm{end}}$, which we call *slow-growing*. We call the latter two outcomes—extinction and slow growth of the tumour cell population—*suppression*. While our system is stochastic, different outcomes may arise among individual realisations under the same parameter set and initial condition. Thus, we measure the proportion of simulations ending in a given outcome under different parameter values.

*Table 1* summarises all notation and parameters governing the model, including baseline values used for all simulations unless otherwise specified.

## Additional information

### Funding

| Funder | Grant reference number | Author |
|---|---|---|
| European Commission | 10.3030/955708 | Christo Morison |

The funders had no role in study design, data collection, and interpretation, or the decision to submit the work for publication.

## Author contributions

Long Wang, Conceptualization, Formal analysis, Validation, Investigation, Visualization, Methodology, Writing – original draft, Writing – review and editing; Christo Morison, Conceptualization, Investigation, Methodology, Writing – original draft, Writing – review and editing; Weini Huang, Conceptualization, Supervision, Funding acquisition, Investigation, Methodology, Writing – original draft, Project administration, Writing – review and editing

## Author ORCIDs

Long Wang (ID) https://orcid.org/0000-0002-6937-2481
Christo Morison (ID) https://orcid.org/0000-0002-9350-7833
Weini Huang (ID) https://orcid.org/0000-0002-9016-2665

Reviewer #1 (Public review): https://doi.org/10.7554/eLife.103970.3.sa1
Author response https://doi.org/10.7554/eLife.103970.3.sa2

# Additional files

## Supplementary files

MDAR checklist

## Data availability

The current manuscript is a computational study, so no data have been generated for this manuscript. Modelling code is available at https://github.com/Bio421/CancerImmuneCoevo (copy archived at *Wang, 2025*).

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

## Appendix 1

This appendix is split into two sections: stochastic analysis and extended figures. We first state theoretical distributions from the literature (section Stochastic analysis). Second, we define the neoantigen burden distribution (NBD) and derive its expected distribution, as well as plot the correlation between the mean of the NBD and the mean of the SFS (section Neoantigen burden distribution). We then discuss early dynamics (section Early stochastic dynamics). Next, we investigate the effector cell population spikes and derive approximations for their expected frequency and amplitudes (section Effector cell population spikes). Finally, we discuss the implications of a faster immune response due to an existing effector pool in the body on the genetic evidence for selection (section Faster immune response). Additional figures of stochastic simulations are found in (sections Composition of effector population and Genetics).

### Stochastic analysis

#### Expected mutational distributions

We begin by stating known expressions for the expected SFS and MBD in the case of neutral evolution, which we used for plots in *Figure 4* and *Figure 3*. From *Gunnarsson et al., 2021*, the expected neutral SFS at large population size $K$ with mutations accumulating at rate $\mu$, conditional on survival (which we do by filtering out the realisations that do not make it to population $K$) is given by

$$\mathbb{E}\left[S_j \,|\, \text{survival}\right] \approx \mu K \sum_{i=0}^{\infty} \frac{\left(\frac{d}{b}\right)^i}{(i+j)(i+j+1)}. \tag{3}$$

We take $\mu = \lambda(1 - p_a)$ for neutral mutations, and $\mu = \lambda p_a$ for antigenic mutations.

Under the same assumptions, the expected neutral MBD can be calculated as follows *Morison et al., 2023*:

$$\mathbb{E}\left[B_k \,|\, \text{survival}\right] \approx \sum_{\ell} D_\ell \frac{\mathrm{e}^{-\ell\mu}(\ell\mu)^k}{k!} \approx \sum_{\ell} \left(\frac{(b+d)\mu \log K}{b-d}\right)^\ell \frac{K^{-\mu(b+d)/(b-d)}}{\ell!} \frac{\mathrm{e}^{-\ell\mu}(\ell\mu)^k}{k!}, \tag{4}$$

where the sum over $\ell$ spans at most the number of events (births or deaths), though the sum converges faster. We have used a continuous-time approximate version from *Kharlamov, 1969* for the expected division distribution $D_\ell$ instead of the discrete-time expression from *Morison et al., 2023* for ease of computation, along with the approximation that the population grows to $K$ in $\log K/(b-d)$ time, which arises by inspection of the deterministic case of exponential growth.

#### Neoantigen burden distribution

The immune system reacts to the antigenicities of the antigenic mutations possessed by a cancer cell rather than simply its (antigenic) mutational burden. To capture this nuance, we define the neoantigen burden $N_\ell$ as the number of cells with cumulative antigenicity $\sum_i A_i$ (where the sum spans the antigenic mutations $i$ within a cell) in $[\ell, \ell+1)$. Taken together, the elements $N_\ell$ form the NBD.

Following the derivation of the MBD from the division distribution in *Morison et al., 2023*, we write $A_{i,j,k} \sim \text{Exp}(1)$ for the antigenicity of the $i$th antigenic mutation of the $j$th cell (for some labelling of cells $1 \leq j \leq B_k$) having a mutational burden of $k$ and $\mathbb{1}_A$ for the indicator function that is 1 on the set $A$ and 0 otherwise. Conditioning on knowledge of the antigenic MBD $\{B_k\}_{k=1}^M$, we find

$$\mathbb{E}\left[N_\ell | \{B_k\}_k\right] = \mathbb{E}\left[\sum_{k=1}^{M}\sum_{j=1}^{B_k} \mathbb{1}_{\left\{\sum_{i=1}^k A_{i,j,k} \in [\ell,\ell+1)\right\}} | \{B_k\}_k\right] = \sum_{k=1}^{M} B_k \mathbb{E}\left[\mathbb{1}_{\left\{\sum_{i=1}^k A_{i,j,k} \in [\ell,\ell+1)\right\}}\right]. \tag{5}$$

A sum of $k$ exponentially distributed random variables with means $\nu$ obeys an Erlang distribution, which has probability density function

$$f(t; k, \nu) = \frac{\nu^k t^{k-1} \mathrm{e}^{-\nu t}}{(k-1)!}. \tag{6}$$

We take expectations of both sides of *Equation 5* (where $\nu = 1$) and use the law of total expectation $\mathbb{E}[X] = \mathbb{E}[\mathbb{E}[X|Y]]$ and *Equation 6* to find

$$\mathbb{E}[N_\ell] = \sum_{k=1}^{M} \mathbb{E}[B_k] \int_\ell^{\ell+1} \frac{t^{k-1}e^{-t}}{(k-1)!}\, dt = e^{-\ell} \sum_{k=1}^{M} \mathbb{E}[B_k] \sum_{k'=0}^{k-1} \frac{\ell^{k'} - e^{-1}(\ell+1)^{k'}}{k'!}. \tag{7}$$

This allows for a flexible definition of the NBD: we have chosen to discretise at integer intervals $[\ell, \ell+1)$, but could have kept it continuous with probability density function $\sum_k \mathbb{E}[B_k]\, t^{k-1}e^{-t}/(k-1)!$.

*Appendix 1—figure 2* depicts the average NBD (along with the corresponding construction for the immunogenicity) over 100 realisations, when $\lambda = 1$ (*Appendix 1—figure 2A, B*) and when $\lambda = 10$ (*Appendix 1—figure 2C, D*). Both low and high immune impact parameter sets are plotted, to demonstrate that much like the MBD (see *Figure 5* and *Appendix 1—figure 13*), the tail of the distribution is shortened in the case of higher selection. This is because cells with higher antigenicities (likewise for those with higher immunogenicities) face stronger negative selection from the effector population.

## Early stochastic dynamics

On average, an effector population of type $i = 1$ will have passive capacity $B/D$: that is, at a population of $E_1 = B/D$, its passive recruitment rate and death rate are equal, and thus ignoring its interactions with its target population $C_1$, $E_1$ will hover around this value. Once $C_1$ grows to $D/(\alpha_1 - \gamma_1)$, however, the active recruitment rate will equal the death rate. We can thus qualitatively describe the early dynamics of a cancer clone as follows: the mutation arises; the effector population grows to $B/D$; the clone grows to $D/(\alpha_1 - \gamma_1)$, at which point the effector population increases to properly (i.e. via active recruitment) combat the threat.

For most parameter choices in the main text, $B/D = O(1)$ and $D/(\alpha_1 - \gamma_1) = O(10^2)$, so that the effector population size only manages to grow to more than a few cells once its targets number around a hundred. It is also worth mentioning that early ordinary differential equation models of tumour–immune dynamics phrased in a predator–prey fashion use Kuznetsov et al.'s estimate of effector passive recruitment being $O(10^4)$ cells per day (*Kuznetsov et al., 1994*). With $O(0.1)$ or $O(1)$ new antigenic mutations (and thus effector types) generated in each division, when our cancer cell population size is $O(10^4)$, we therefore expect a passive recruitment similar to *Kuznetsov et al., 1994*.

Models of immune responses vary in their triggers: Chen et al. suppose a minimal proportion of antigenic cancer cells (*Chen et al., 2024*), the growth threshold conjecture requires a minimal cancer growth rate (*Arias et al., 2015*), and most ordinary differential equation models as well as Lakatos et al. suppose instantaneous responses (*Lakatos et al., 2020*; *d'Onofrio, 2005*). While our mechanistic model does not assume frequency dependence, the previous discussion implies that the immune response implicitly requires a certain cancer density before it can grow effectively to fight.

## Effector cell population spikes

As identified in individual-based simulations (see *Figure 2*, for instance), effector subpopulations in the stochastic model can undergo rapid spiking events, before vanishing to zero when the targeted mutant (labelled by $i$) goes extinct (and thus the effector cells have no more prey and so are removed from the system). These spikes arise when the mutation has immunogenicity $I_i$ much higher than the mean $\langle I \rangle$, as well as an antigenicity $A_i$ lower than the mean $\langle A \rangle$, as otherwise the spike would not need to grow since the effector cells would be more efficient at killing. Qualitatively, we will write $\hat{I}$ ($\hat{A}$) for the threshold immunogenicity (antigenicity) and above (below) which spikes may occur.

Effector spikes arise when the population sizes enter an orbit that ends when $C_i(t) < 1$ (that is, extinction of mutation $i$). We write $i = 1$ for the mutation causes the spike, since we consider it to be arising in a neutral cell. (We argue qualitatively that were other antigenic mutations carried by the cell to generate an effector spike, they already would have; thus we expect most spikes to arise in neutral or almost neutral cells.)

Using that immunogenicities and antigenicities are drawn from exponential distributions with means 1, the rate of spikes $r_{\text{spike}}$ is then

$$r_{\text{spike}} = p_a \lambda e^{-(\hat{I}+\hat{A})}, \tag{8}$$

since the rate of new antigenic mutations and the probabilities $\mathbb{P}(I_i > \hat{I})$ and $\mathbb{P}(A_i > \hat{A})$ are independent.

## Faster immune response

Our immune system possesses an innate pool of effector cells, able to respond to threats without the time-consuming training assumed implicitly in our specialised model of active and passive recruitment (*Chen and Mellman, 2013*). With some probability, this pool might include effector cells that can recognise newly arising neoantigens quickly. Thus, rather than having to wait for the first passively recruited effector cell, the body could react faster to novel antigenic mutations.

To investigate the impact of this mechanism on the observed coevolution, we modified our model to include an effector cell of type $i$ upon initiation of antigenic mutation $i$. *Appendix 1—figure 1A, B* shows the average number of antigenic mutations per cell and *Appendix 1—figure 1C, D* shows the SFS, in the case of $\lambda = 1$.

When comparing with the corresponding figures for the model used throughout the main text (*Figure 4A, B*, *Figure 5B*, and *Figure 5F*, respectively), the patterns appear the same, though more evident. This reinforces that the speed of the immune response increases its effectiveness, though it does not change the qualitative nature of cancer–immune coevolution. However, rather than having a guaranteed effector cell that can handle a newly arising neoantigen, there is some probability of the effector pool containing the desired specialty. Thus, we expect that in the body, selection will follow the patterns shown in both cases (*Figures 4 and 5* vs. *Appendix 1—figure 1*), at some magnitude between the two, related to the aforementioned probability.

## Extended figures

### Composition of effector population

*Appendix 1—figure 3* illustrates the temporal dynamics of the composition of the effector cell population for the same two stochastic realisations shown in *Figure 2*. In the case of low mutation rate ($\lambda = 1$, *Figure 2A*), the emergence of effector cells is delayed, and a single effector cell type dominates the population and then goes extinct after eliminating all antigenic targets. New types of effector cells arise again later when new antigenic mutations arise in cancer cells. In contrast, under high mutation rates ($\lambda = 10$, *Figure 2B*), the effector cells arise quickly in the start of the simulation and often multiple effector cell types dominate the population simultaneously.

When considered alongside *Figure 2*, these Muller plots demonstrate that spikes in the dynamics of the average antigenicity and immunogenicity are consistently associated with the expansion of specific dominating effector cell types in response to the presence of their specific antigenic targets.

### Cyclic dynamics

*Appendix 1—figure 4* shows the phase trajectories of cancer and effector cell abundances for the same two realisations presented in *Figure 2*. Unlike classical predator–prey models, which often exhibit periodic dynamics, the phase trajectories of this system display more stochastic and irregular behaviour.

To quantify the cyclic dynamics of our simulations, we developed a method to enumerate the number of cycles present in each realisation. We track the directional changes in the phase portrait and accumulate the angular displacement. When the cumulative direction change exceeds $2\pi$ radians (anti-clockwise) or $-2\pi$ radians (clockwise), we record this as one complete cycle. To validate the robustness of this approach, we applied the same methodology to a non-evolving predator–prey model with stochastic simulations, where the dynamics are relatively periodic. In short, the available reactions in the stochastic system are as follows:

$$
\begin{aligned}
(\text{prey}) &\xrightarrow{\text{prey birth}} (\text{prey}), (\text{prey}) \\
(\text{prey}) &\xrightarrow{\text{prey death}} \varnothing \\
(\text{predator}) &\xrightarrow{\text{predator death}} \varnothing \\
(\text{prey}), (\text{predator}) &\xrightarrow{\text{predation}} (\text{predator}) \\
(\text{prey}), (\text{predator}) &\xrightarrow{\text{predator reproduction}} (\text{prey}), (\text{predator}), (\text{predator}),
\end{aligned}
\tag{9}
$$

with the classical Lotka–Volterra predator–prey equation arising in the deterministic limit. With this, we observed mostly the same number of cycles across realisation (see *Appendix 1—figure 5*).

Applying this method, we obtain the distribution of cycle counts in our system shown in *Appendix 1—figure 6*. For the low mutation rate case ($\lambda = 1$), the majority of realisations exhibit no cycles, whereas for the hyper-mutated case ($\lambda = 10$), the number of cycles typically ranges from 1 to 2, with only a few realisations displaying more than 2 cycles within our simulation time. These cyclic dynamics are driven by the expansion of specific antigenic mutations that possess favourable characteristics: relatively high immunogenicity enabling substantial effector cell recruitment, coupled with relatively low antigenicity that prevents rapid elimination by the immune response (see *Figure 2* and *Appendix 1—figure 3*, as well as Appendix Section Effector cell population spikes). We further investigated the occurrence of irregular (clockwise) cycles, which are atypical in antagonistic interactions but possible under coevolution. As shown in *Appendix 1—figure 7*, approximately 10% of realisations in the high mutation rate case exhibit one or two irregular cycles. This suggests that while the system can exhibit irregular cyclic behaviour under certain conditions, such dynamics do not represent the predominant mode of behaviour.

## Population dynamics

*Appendix 1—figure 8* shows representative population dynamics (first row), along with the corresponding composition of the tumour population (second row), when $\lambda = 1$ (*Appendix 1—figure 8A–D*) and $\lambda = 10$ (*Appendix 1—figure 8E–J*). In the upper row, a blue histogram depicts the end times for the realisations, along with the average time $T_K$ to reach population size $K = 3 \times 10^4$, when applicable. (As seen in *Appendix 1—figure 8I*, all realisations go extinct, so no such $T_K$ is defined.) The red lines in the second row refer to the number of allowed antigenic mutations possessed by a cancer cell to consider it neutral. That is, the line labelled by three means that cells with three antigenic mutations are deemed neutral, whereas those with four antigenic mutations are considered antigenic and thus contribute to the proportion of the tumour that is antigenic. Note that since the immunogenicities and antigenicities are drawn from exponential distributions with mode 0, it is likely that many purportedly antigenic mutations $i$ have very weak immunogenicity $I_i \approx 0$ and antigenicity $A_i \approx 0$. In this case, they would interact very rarely with the immune system (being effectively neutral), while still contributing to a count of cancer cells carrying antigenic mutations.

In *Appendix 1—figure 8*, the stopping time for the realisations $T_{\text{end}}$ was made large so as to establish a good estimate on the average time $T_K$ that the cancer cell population (in the non-suppressed case) reaches the stopping population size $K$. Due to the stochasticity of the model, there is a distribution over end times for each realisation even for a single set of parameters. It is clear from the difference in values of the mean $T_K$ for *Appendix 1—figure 8A, C* (likewise for *Appendix 1—figure 8E, G*) that the immune system plays a role in slowing the cancer growth, since avg $T_K^{\mathbf{A}} <$ avg $T_K^{\mathbf{C}}$ (and avg $T_K^{\mathbf{E}} <$ avg $T_K^{\mathbf{G}}$).

## Heat maps

*Appendix 1—figure 9A, B* depicts the extinction times for simulations run with $\lambda = 1$ and $\lambda = 10$, respectively. *Appendix 1—figure 10A–C and D–F* depicts what proportion of tumour cells carry antigenic mutations for simulations run with $\lambda = 1$ and $\lambda = 10$, respectively. The first row allows no antigenic mutations in neutral cancer cells; the second (third) row allows one (two) antigenic mutation(s) before a cell is considered antigenic.

## Genetics

When the immune system is absent, the SFS and MBD are well approximated by the theoretical predictions (black dashed lines; see *Appendix 1—figure 11*). In the presence of an immune system, however, the MBDs deviate from their theoretical predictions, as expected. When hyper-mutated tumours are considered ($\lambda = 10$), the genetic footprints of selection are more pronounced. *Appendix 1—figure 12* depicts the average number of antigenic mutations per cancer cell, and *Appendix 1—figure 13* shows the SFS and MBD in this regime.

To validate the sensitivity of the Wasserstein distance to interaction parameters, we plot a heat map (*Appendix 1—figure 14*), which shows the similarity pattern with *Figure 3* that with the increase of interaction parameters, the MBD deviates from the theoretical predictions as expected.

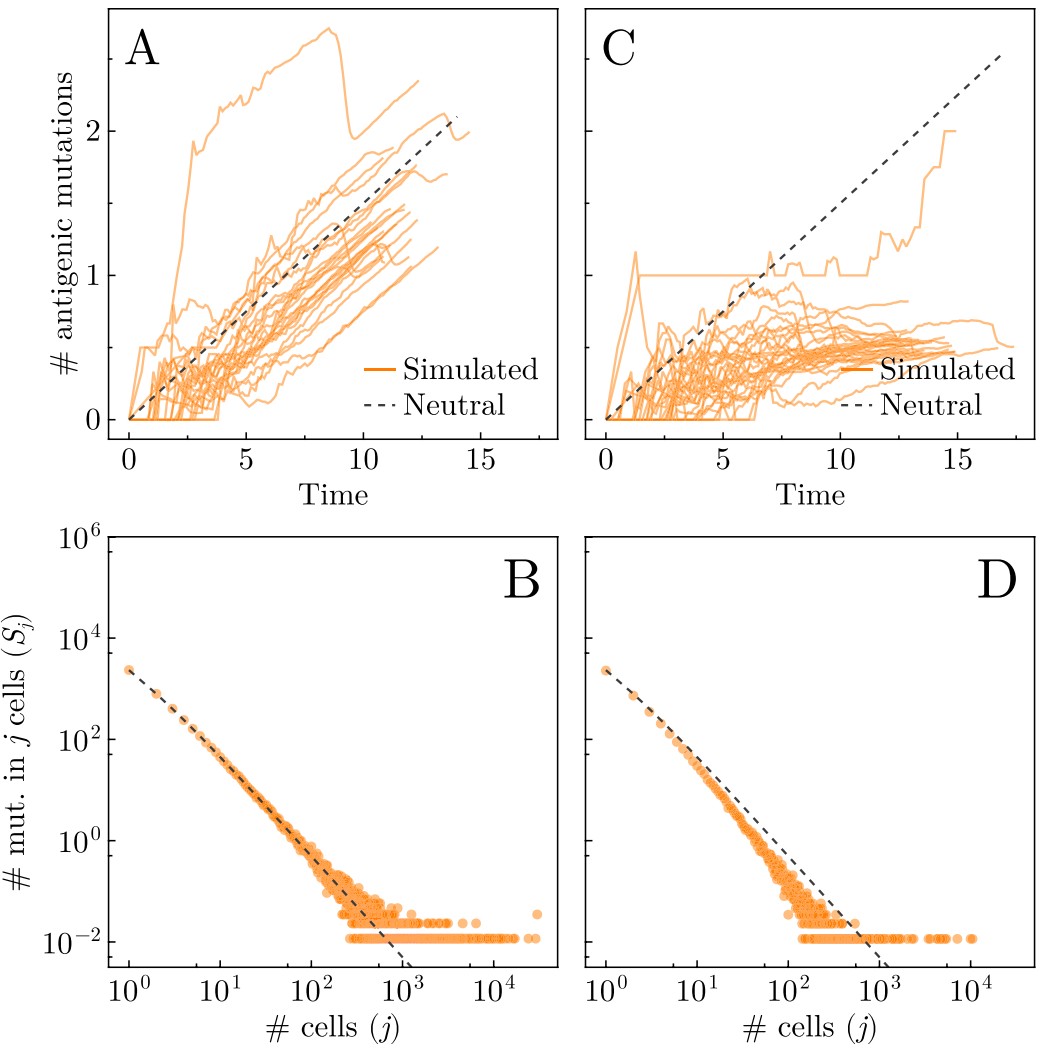

**Appendix 1—figure 1.** Genetic evidence of selection for a faster-acting immune system. (**A, B**) Average number of antigenic mutations per cell in solid orange lines for several representative realisations when $\lambda = 1$. Theoretical prediction for the accumulation of neutral mutations per cell in an exponentially growing population shown in grey dashed line. (**C, D**) Simulated SFS for antigenic mutations averaged over 100 realisations when $\lambda = 10$ (orange points), along with the theoretical predictions (black dashed lines) in the absence of an immune response. (**A, C**) Low immune effect: $\alpha_0 = 0.002$ and $\beta_0 = 0.001$. (**B, C**) High immune effect: $\alpha_0 = 0.03$ and $\beta_0 = 0.3$. (Parameter sets chosen as points $a$ and $b$ from **Figure 3A**).

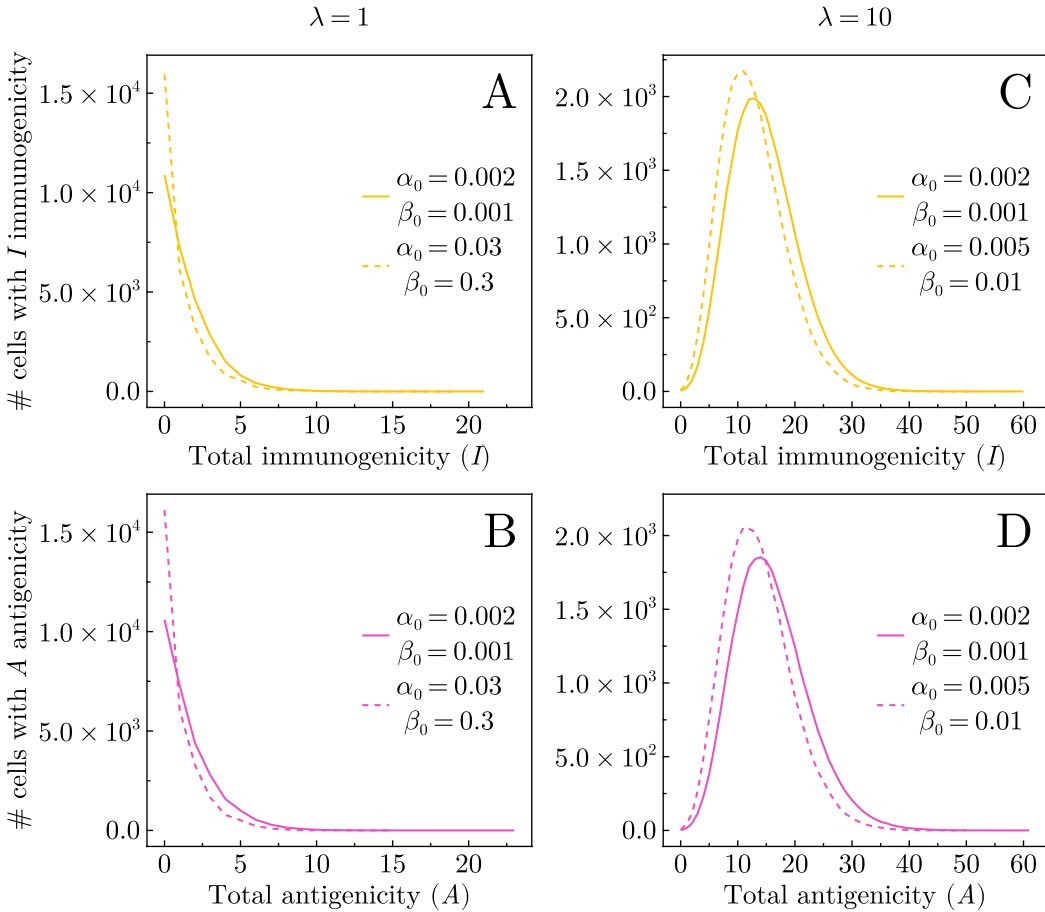

**Appendix 1—figure 2.** Neoantigen burden distribution (NBD) (yellow lines) and the corresponding distribution for immunogenicity (purple lines) averaged over 100 realisations, when $\lambda = 1$ (**A, B**) and $\lambda = 10$ (**C, D**), for both low (dotted lines) and high (solid lines) immune effectiveness. All parameter values not specified here are listed in *Table 1* of the main text.

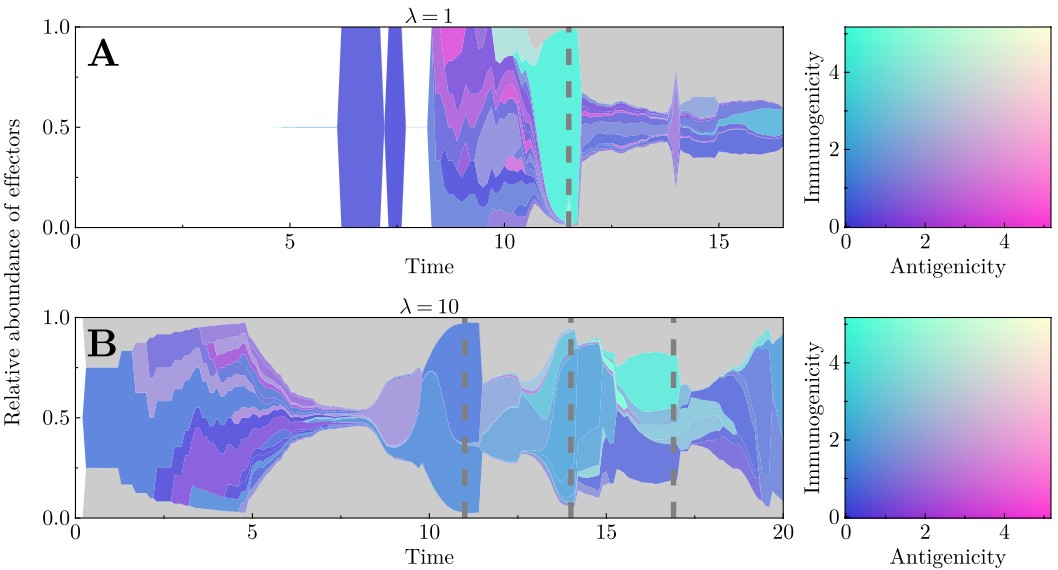

**Appendix 1—figure 3.** Muller plots for two representative realisations from *Figure 2*. (**A**, **B**) correspond to *Figure 2A, B and C, D*, respectively. Vertical dashed grey lines in all panels refer to the timing of spikes in the effector population in *Figure 2*, also indicated by vertical dashed grey lines. The colour indicates the antigenicity and immunogenicity of the antigen corresponding to each effector type, as shown in the colour maps on the right. White regions indicate the absence of effector cells, while grey represents all rare effector types whose abundance remains below 5%. Interaction parameters: $\alpha_0 = 0.03$, $\beta_0 = 0.3$ (**A**); $\alpha_0 = 0.005$, $\beta_0 = 0.01$ (**B**).

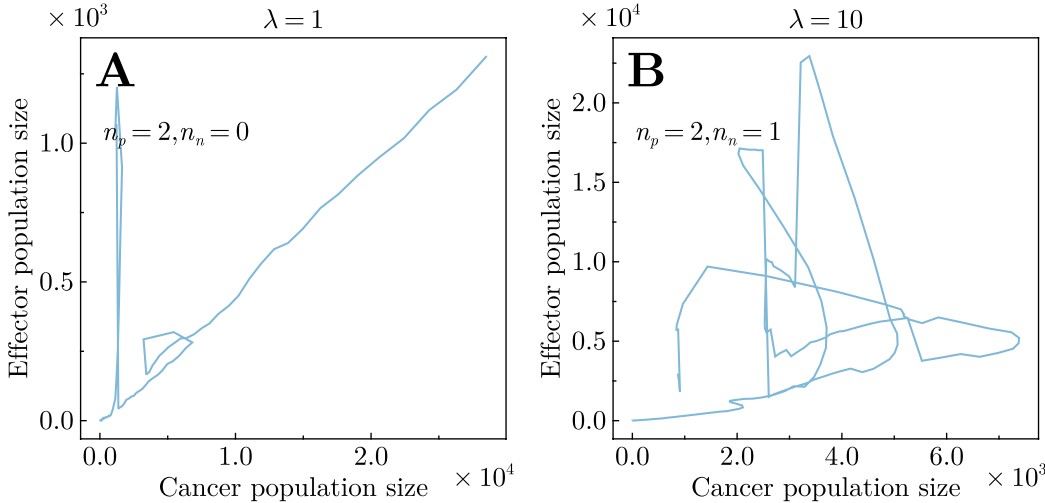

**Appendix 1—figure 4.** Phase trajectories of two realisations in *Figure 2*. (**A**, **B**) are corresponding to *Figure 2A, B and C, D*, respectively. Interaction parameters: $\alpha_0 = 0.03$, $\beta_0 = 0.3$ (**A**); $\alpha_0 = 0.005$, $\beta_0 = 0.01$ (**B**).

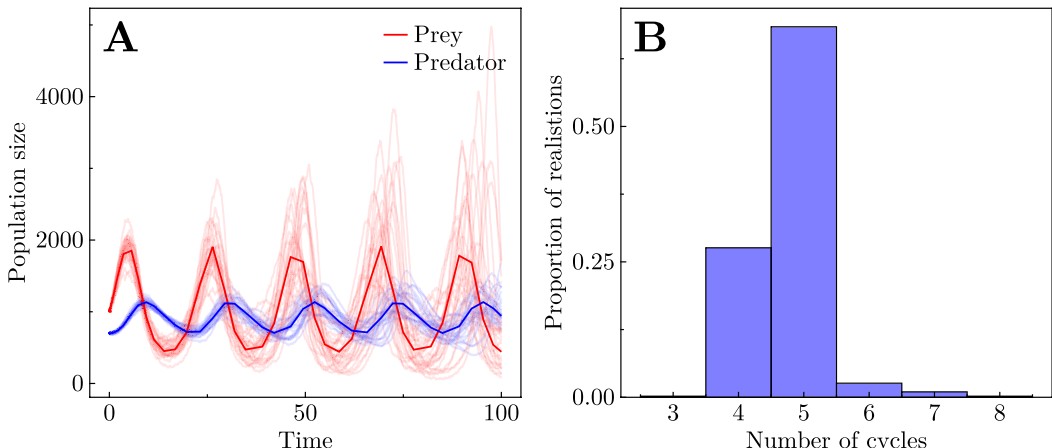

**Appendix 1—figure 5.** Validation of our cycle counting method in a non-evolving prey–predator system. (**A**) Population dynamics of the deterministic system (dark lines) with 20 stochastic realisations (pale lines). (**B**) Inferred number of predator–prey cycles using the same methodology as in *Appendix 1—figure 6*. Model parameters, from *Equation 9*: prey birth rate = 1, prey death rate = 0.1, predator death rate = 0.1, predation rate = 0.001, and predator reproduction rate = 0.0001. Initial population sizes of the prey and the predator were 1000 and 700, respectively, with a maximum simulation time of 100. The histogram is constructed from 500 independent realisations.

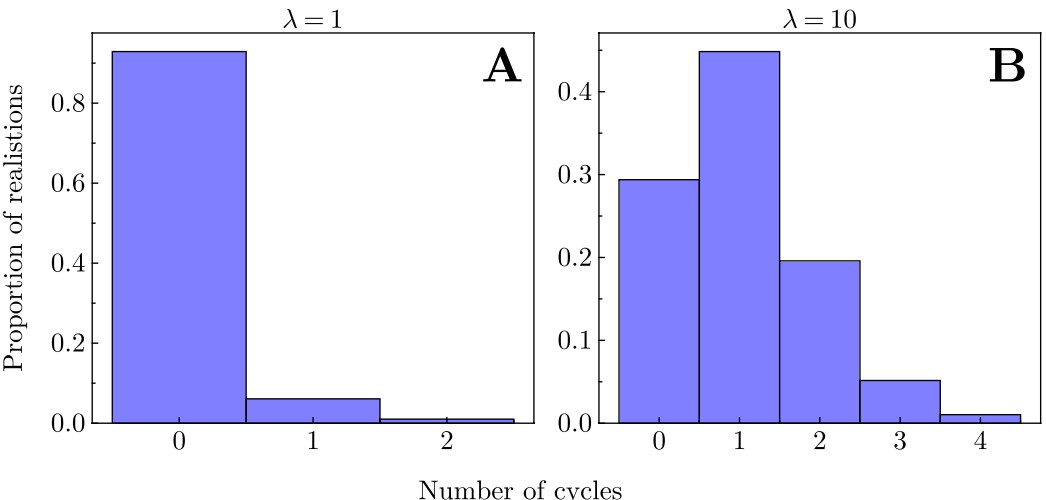

**Appendix 1—figure 6.** Histograms depicting the inferred number of cycles in the population dynamics. (**A, B**) Distribution of cycle counts for simulations with, respectively, low ($\lambda = 1$) and high ($\lambda = 10$) mutation rates. Each histogram is constructed from 200 independent realisations. The interaction parameters are $\alpha_0 = 0.03$ and $\beta_0 = 0.3$ (**A**); $\alpha_0 = 0.005$ and $\beta_0 = 0.01$ (**B**).

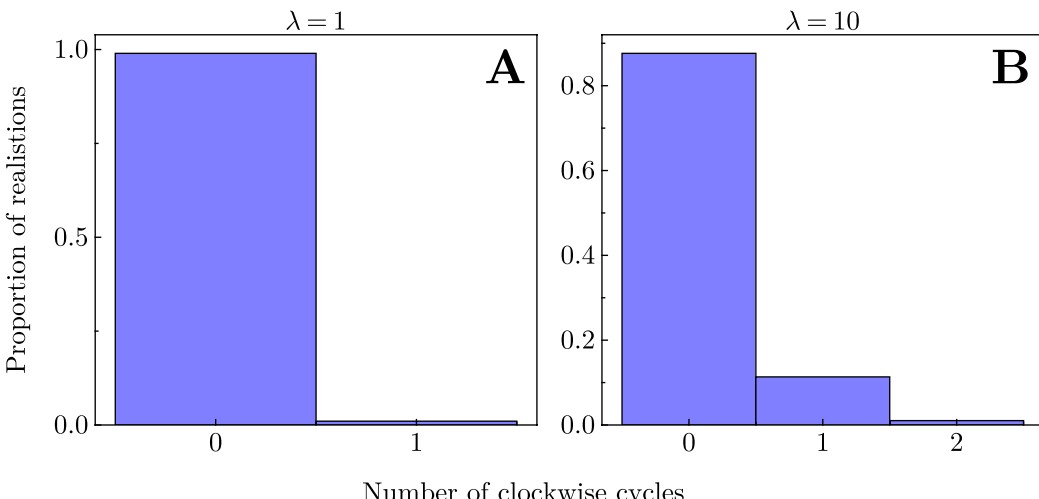

**Appendix 1—figure 7.** Histograms depicting the inferred number of irregular (clockwise) cycles in the population dynamics. (**A, B**) Distribution of cycle counts for simulations with, respectively, low ($\lambda = 1$) and high ($\lambda = 10$) mutation rates. Each histogram is constructed from 200 independent realisations. The interaction parameters are $\alpha_0 = 0.03$ and $\beta_0 = 0.3$ (**A**); $\alpha_0 = 0.005$ and $\beta_0 = 0.01$ (**B**).

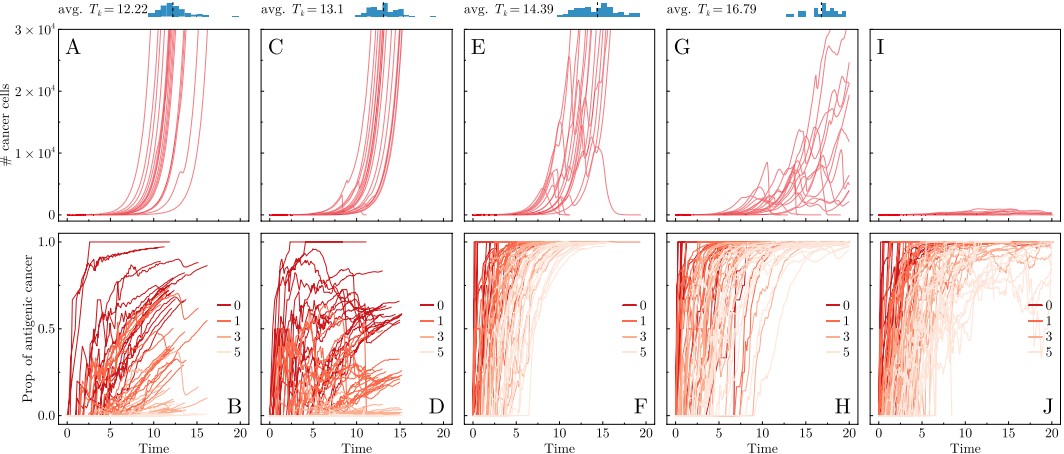

**Appendix 1—figure 8.** Cancer–effector population dynamics (first row) and tumour antigenicity (second row) for five different sets of parameters. (**A, B**) $\lambda = 1$, $\alpha_0 = 0.002$ and $\beta_0 = 0.001$. (**C, D**) $\lambda = 1$, $\alpha_0 = 0.03$ and $\beta_0 = 0.3$. (**E, F**) $\lambda = 10$, $\alpha_0 = 0.002$ and $\beta_0 = 0.001$. (**G, H**) $\lambda = 10$, $\alpha_0 = 0.005$ and $\beta_0 = 0.01$. (**I, J**) $\lambda = 10$, $\alpha_0 = 0.03$ and $\beta_0 = 0.3$. In the first row, red lines depict cancer cell population sizes for several representative realisations. Blue histograms above show the final time $T_K$ where the population size reached $K = 3 \times 10^4$ for all non-extinct realisations, with a mean given by a dashed vertical black line. In the second row, red lines with increasing paleness, labelled by $i = 0$, 1, 3, or 5, depict the proportion of cancer cells contain $i$ or more antigenic mutations. All parameter values not specified here are listed in **Table 1** of the main text.

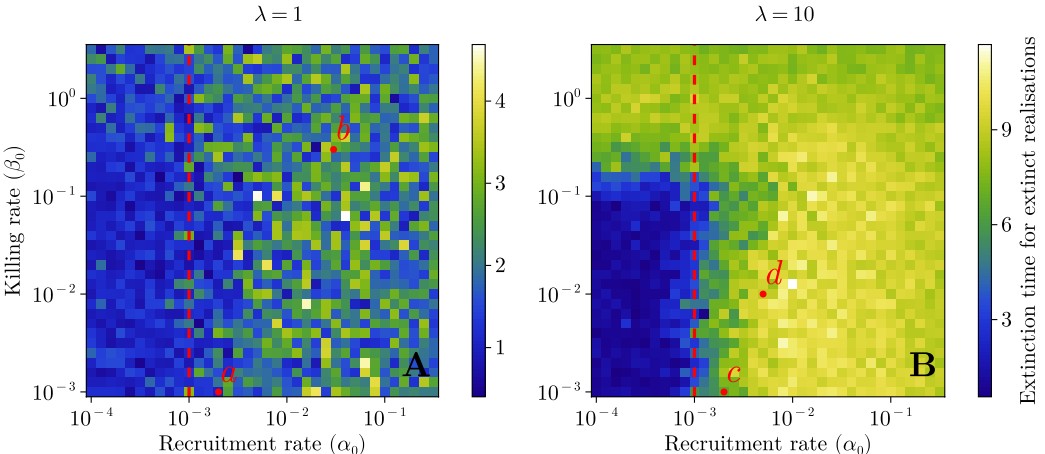

**Appendix 1—figure 9.** Extinction time heat maps for $\lambda = 1$ (**A**) and $\lambda = 10$ (**B**). All parameter values not specified here are listed in **Table 1** of the main text.

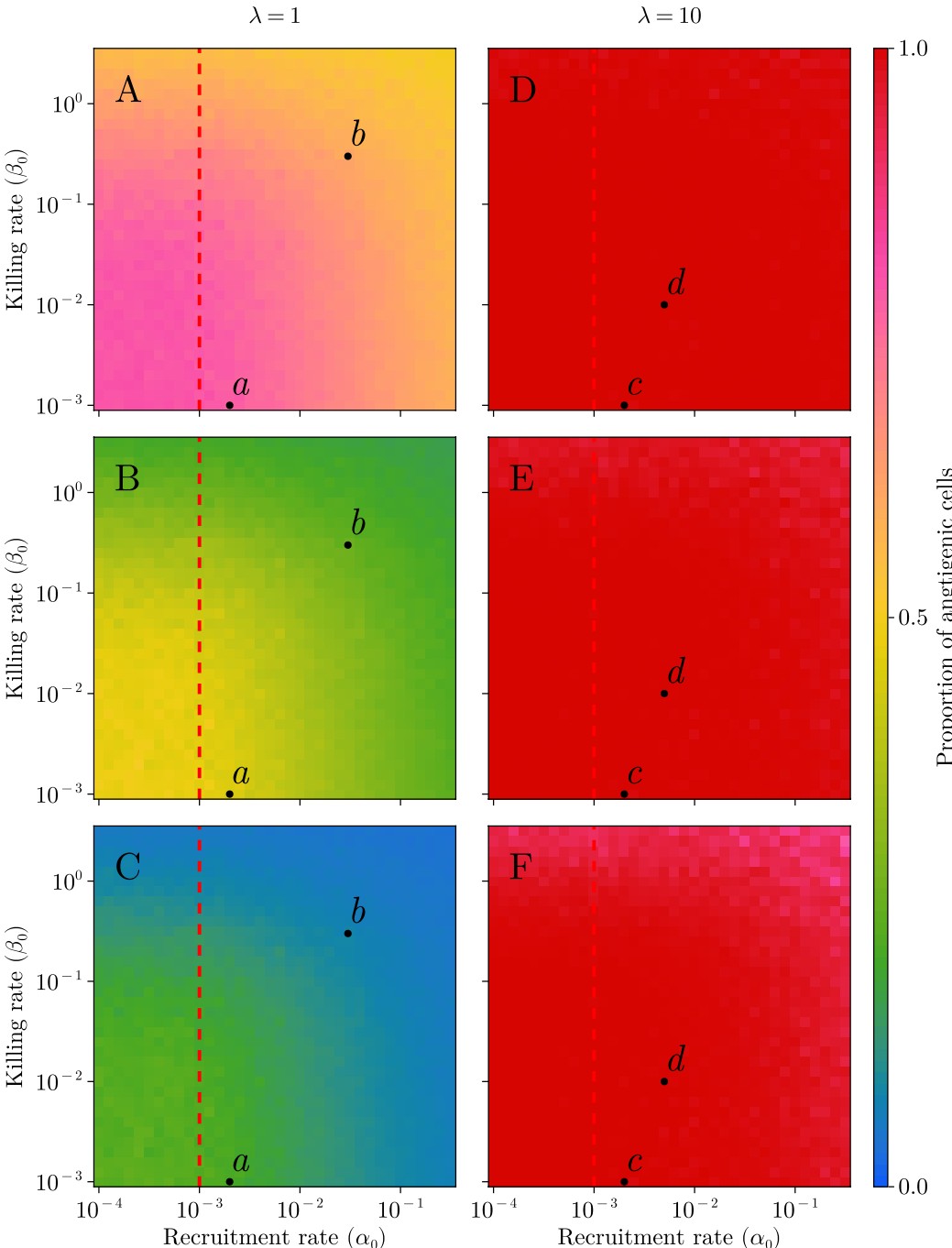

**Appendix 1—figure 10.** Heat maps depicting the tumour composition (that is, the proportion of tumour cells that are antigenic) for $\lambda = 1$ (**A—C**) and $\lambda = 10$ (**D—F**). The first row (**A & D**) is where one antigenic mutation held by a cell makes the cell antigenic; the second (**B & E**) and third rows (**C & F**) allow for one and two antigenic mutations (respectively) to be possessed by a cancer cell while still considering it neutral. Points $a$ and $c$ (respectively, $b$ and $d$) label parameter sets of low (respectively, high) immune effectiveness. All parameter values not specified here are listed in **Table 1** of the main text.

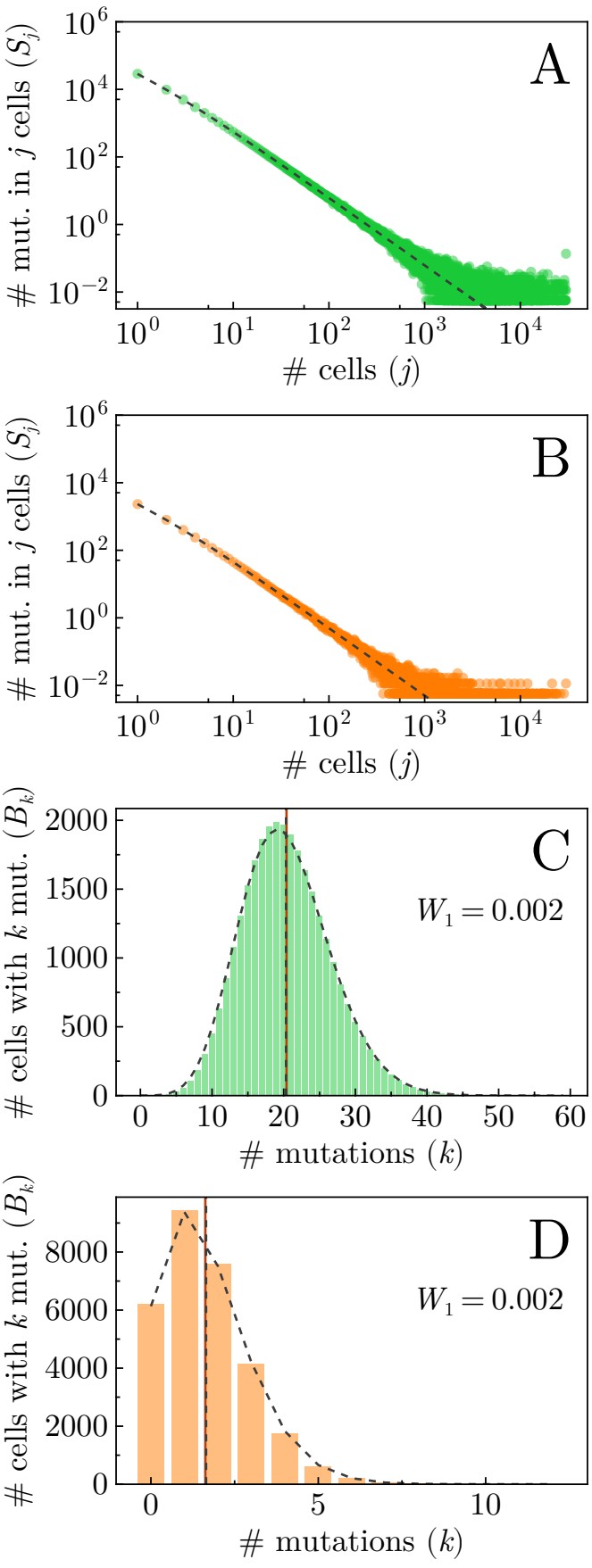

**Appendix 1—figure 11.** Genetic markers of selection: SFS (**A, B**) and MBD (**C, D**) for no immune effect case (all interaction parameters set to 0) averaged over 100 realisations when $\lambda = 1$, along with the theoretical predictions (black dashed lines) in the absence of an immune response. Green data represents neutral mutations (**A, C**) and orange data represents antigenic mutations (**B, D**), with vertical lines representing the means of the distributions for MBDs in panels (**C, D**). All parameter values not specified here are listed in **Table 1** of the main text.

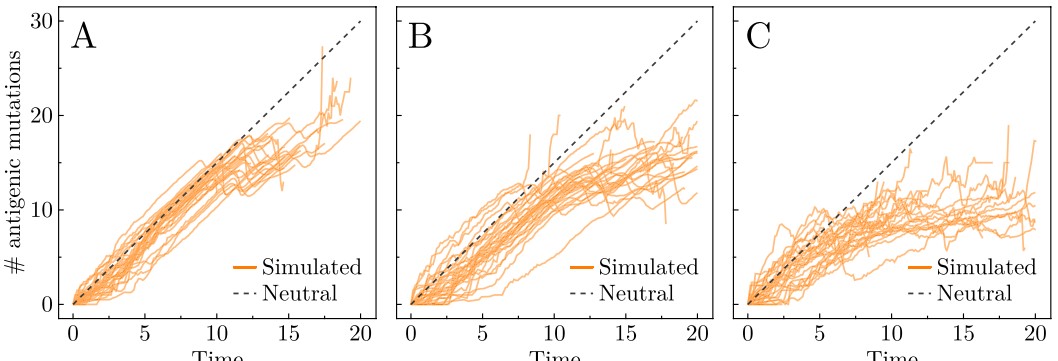

**Appendix 1—figure 12.** Average number of antigenic mutations in solid orange lines for several representative realisations when $\lambda = 10$. Theoretical prediction for the accumulation of neutral mutations in an exponentially growing population is shown in grey dashed line. (**A**) Low immune effect: $\alpha_0 = 0.002$ and $\beta_0 = 0.001$. (**B**) Middling immune effect: $\alpha_0 = 0.005$ and $\beta_0 = 0.01$. (**C**) High immune effect: $\alpha_0 = 0.03$ and $\beta_0 = 0.3$. (Parameter sets for (**A**) and (**C**) chosen as points $c$ and $d$ from **Figure 3C**, respectively.)

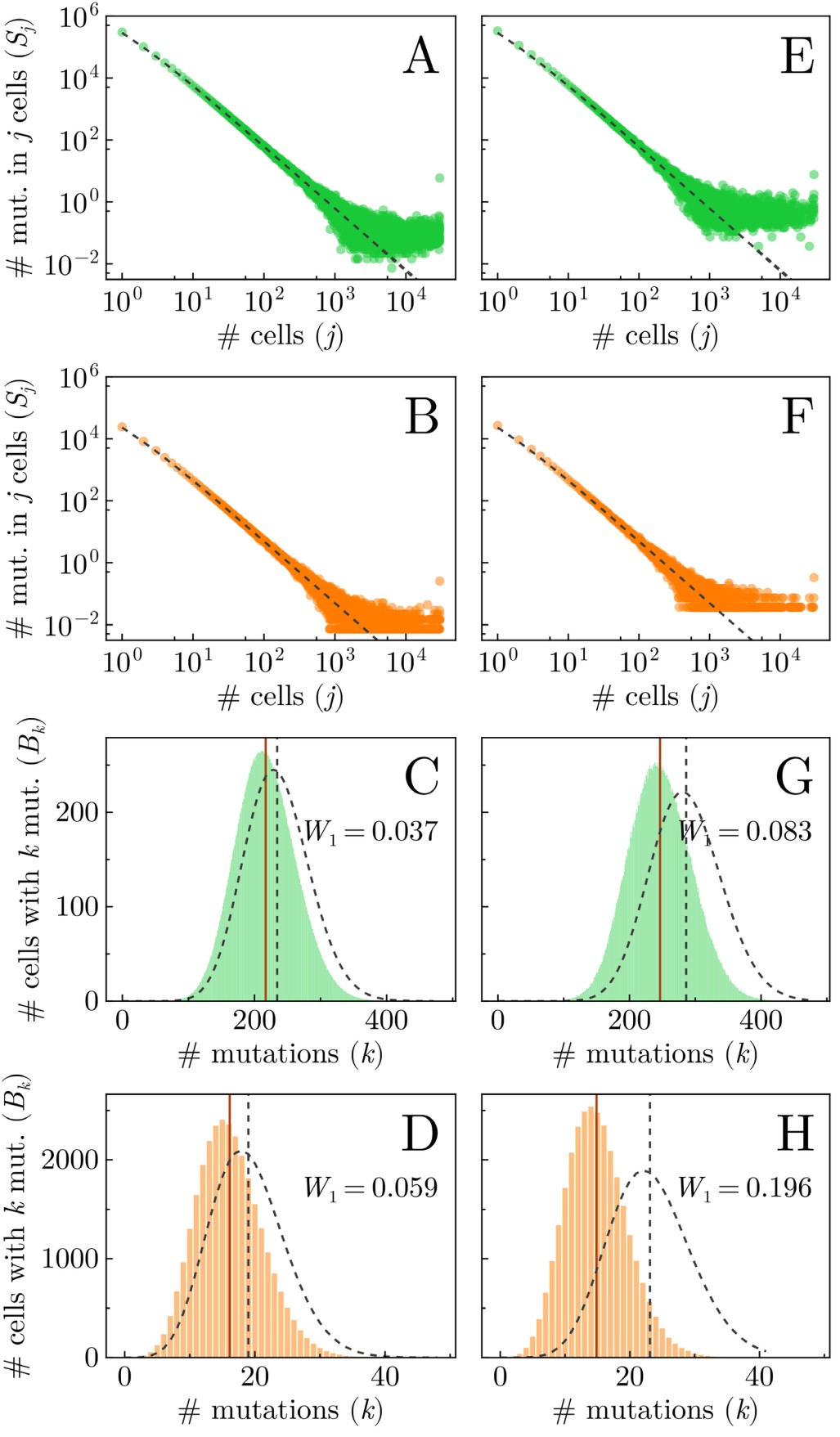

**Appendix 1—figure 13.** Genetic markers of selection: SFS (**A, B, E, F**) and MBD (**C, D, G, H**) for low (left column; $\alpha_0 = 0.002$ and $\beta_0 = 0.001$, point c in **Figure 3**) and high (right column; $\alpha_0 = 0.005$ and $\beta_0 = 0.01$, point d in **Figure 3**) immune effectiveness averaged over 100 realisations when $\lambda = 10$, along with the theoretical predictions (black dashed lines) in the absence of an immune response. Green data represents neutral mutations (**A, C, E, G**) and orange data represents antigenic mutations (**B, D, F, H**), with vertical lines representing the means of the distributions for MBDs in panels (**C, D, G, H**). All parameter values not specified here are listed in **Table 1** of the main text.

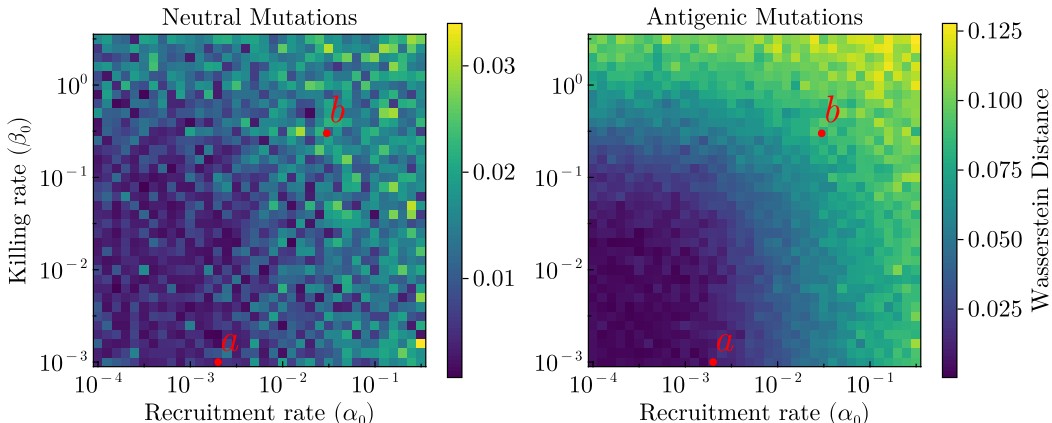

**Appendix 1—figure 14.** Heat maps illustrate the differences between the simulated and theoretical single-cell mutation burden distributions when $\lambda = 1$. The colour gradient represents the Wasserstein distance, with the number of mutations $k$ rescaled to the interval [0,1]. The left panel displays how the difference between simulated and theoretical predicted neutral MBDs varies with interaction parameters, while the right panel shows the same for antigenic mutations. Point a and b are the same in **Figure 3** and all other parameter values can be found in **Table 1** in the main text.

