## [Editor Report · eLife Assessment]

This **important** work presents a stochastic branching process model of tumour-immune coevolution, incorporating stochastic antigenic mutation accumulation and escape within the cancer cell population. They then used this model to investigate how tumour-immune interactions influence tumour outcome and the summary statistics of sequencing data of bulk and single-cell sequencing of a tumour. The evidence is **compelling** and the work will be of interest to cancer-immune biology fields.

---

## [Referee Report · Reviewer #1 (Public review)]

Summary:

The topic of tumor-immune co-evolution is an important, understudied topic with, as the authors noted, a general dearth of good models in this space. The authors have made important progress on the topic by introduced a stochastic branching process model of antigenicity / immunogenicity and measuring the proportion of simulated tumors which go extinct. The model is extensively explored and authors provide some nice theoretical results in addition to simulated results, including an analysis of increasing cancer/immune versus cyclical cancer/immune dynamics. The analysis appropriately builds upon the foundation of other work in the field of predicting site frequency spectrum, but extends the results into cancer-immune co-evolution in an intuitive computational framework.

---

## [Author Response]

The following is the authors’ response to the original reviews.

**Reviewer #1 (Public review):**
Summary:The topic of tumor-immune co-evolution is an important, understudied topic with, as the authors noted, a general dearth of good models in this space. The authors have made important progress on the topic by introducing a stochastic branching process model of antigenicity/immunogenicity and measuring the proportion of simulated tumors that go extinct. The model is extensively explored, and the authors provide some nice theoretical results in addition to simulated results.

We thank the reviewer for the positive comments on our work.

Major commentsThe text in lines 183-191 is intuitively and nicely explained. However, I am not sure all of it follows from the figure panels in Figure 2. For example, the authors refer to a mutation that has a large immunogenicity, but it's not shown how many mutations, or the relative size of the mutations in Figure 2. The same comment holds true for the claim that spikes also arise for mutations with low antigenicity.

We thank the reviewer for helping us to further specify this statement in our original submission. We now added muller plots in a new Appendix Figure (Figure A3) presenting the relative abundances of different types of effector cells in the population over time. Each effector type is colour-coded with its antigenicity and immunogenicity. To align with this Appendix Figure (Figure A3), we also updated our Figure 2 generated under the same realisation as Figure A3. We can now see clearly that the spikes in the mean values of the antigenicity and immunogenicity over the whole effector populations in new Figure 2B&2D indeed correspond to the expansion of single or several antigenic mutations recruiting the specific effector cell types. For example, in Figure 2B, we can see that the spikes of low average antigenicity and high immunogenicity (around time 11) happen at the same time when an effector type in Figure A3 with such a trait (coloured in green) arises and takes over the population. We have rewritten our Results section related (Line 192 - Line 222 in main text and Appendix A6).

**Reviewer #2 (Public review):**
Summary:In this work, the authors developed a model of tumour-immune dynamics, incorporating stochastic antigenic mutation accumulation and escape within the cancer cell population. They then used this model to investigate how tumour-immune interactions influence tumour outcome and summary statistics of sequencing data.Strengths:This novel modeling framework addresses an important and timely topic. The authors consider the useful question of how bulk and single-cell sequencing may provide insights into the tumourimmune interactions and selection processes.

We thank the reviewer for the positive comments.

Weaknesses:One set of conclusions presented in the paper is the presence of cyclic dynamics between effector/cancer cells, antigenicity, and immunogenicity. However, these conclusions are supported in the manuscript by two sample trajectories of stochastic simulations, and these provide mixed support for the conclusions (i.e. the phasing asynchrony described in the text does not seem to apply to Figure 2C).

We have now developed a method to quantify the cyclic dynamics in our system (Appendix A7), where can track the directional changes phase portrait of the abundances of the cancer and effector cells. We first tested this method in a non-evolving stochastic predator-prey system, where our method can correctly capture the number of cycles in this system (Figure A7). We then use this method to quantify the number of cycles we observed between cancer and effector cells under different mutation rates (Figure A5) as well as whether they are counter-clockwise or clockwise cycles (Figure A6). Our results showed that the cyclic dynamics are more often to be observed when mutation rates are higher, and the majority of those cycles are counter-clockwise. When the mutation rate is high, we observe an increase of clockwise cycles, which have been observed in predator-prey systems and explained through coevolution. However, even under high mutation rates, counter-clockwise cycles are still the more frequent type.

In our simulations, we observed rarely out-of-phase cycles, which was by chance present in our original Figure 2. We have now removed that statement about out-of-phase cycles and replaced by more systematic analysis of the cyclic dynamics as described above (Line 192 to 207 in the revised version). We thank the constructive comment of the reviewer, which motivated us to improve our analysis significantly.

Similarly, the authors also find immune selection effects on the shape of the mutational burden in Figure 5 D/H using a qualitative comparison between the distributions and theoretical predictions in the absence of immune response. However the discrepancy appears quite small in panel D, and there are no quantitative comparisons provided to evaluate the significance. An analysis of the robustness of all the conclusions to parameter variation is missing.

We have now added statistical analysis using Wasserstein distance between the simulated mutation burden distribution and theoretical (neutral) expectation in Figure 5 C, D, G, H as well as in Figure A11 C&D when there is no cancer-immune interaction. We can see that the measurements of the Wasserstein distance agrees with our statement, that the higher immune effectiveness leads to larger deviation from the neutral expectation.

Lastly, the role of the Appendix results in the main messages of the paper is unclear.

We agree with the review and have now removed the Appendix sections “Deterministic Analysis”.

Reviewing Editor Comments:I find the abstract too long. For example, "Knowledge of this coevolutionary system and the selection taking place within it can help us understand tumour-immune dynamics both during tumorigenesis but also when treatments such as immunotherapies are applied." can be shortened to: "Knowledge of this coevolutionary system can help us understand tumour-immune dynamics both during tumorigenesis and during immunotherapy treatments."

We agree and have taken the suggestion of the reviewer to shorten our abstract.

**Reviewer #1 (Recommendations for the authors):**
The discussion at lines 134-140, centered around Figure A1, is an important and nicely constructed feature of the model.
**Reviewer #2 (Recommendations for the authors):**
I suggest that the authors conduct a more in-depth analysis of their conclusions on cyclic dynamics over a large set of sample paths.

Done and please see our detailed response to the reviewer 2 above.

In addition, statistical comparisons between the observed mutational burden distribution and theoretical predictions in the absence of immune selection should be carried out to support their conclusions. In all cases, conclusions should be tested extensively for robustness/sensitivity to parameters.

Done and please see our detailed response to the reviewer 2 above.

Here are some specific suggestions/comments:(1) Please provide a precise mathematical description of the model to complement Figure 1.

We have significantly revised our “Model” section to provide a precise mathematical description of our model (Line 138 - 148). Please also see our document showing the difference between the revised version and original submission.

(2) Section on "Interactions dictate outcome of tumour progress" and Figure 3: please define 'tumour outcome' - are the heatmaps produced in Figure 3 tumor size reflecting whether or not the population has reached level K before a particular time? Also, I do not see a definition for the 'slowgrowing' tumour proportion plotted in Figure 3CF or in the accompanying text.

We have now added the definition of “tumour outcome” in our “Model” section (line 171 to 176), where we explain our model parameters and quantities measured in the following “Results” section.

(3) Figure 5C/G: the green dotted vertical line is difficult to see.

We have now changed the mean of the simulations to solid red lines instead of using the green dotted vertical lines previously.

(4) Appendix A1 text under (A2) should U/N be U/C? N does not appear to be defined.

We have more removed the previous A1 section. Please see our response to reviewer 2 as well.

(5) Text under (A5): it is unclear what is meant by "SFS must be heavy tailed (that is, more heterogeneous)" -- a more precise statement regarding tail decay rate and associated consequences would be more helpful.

We have more removed the previous A section, where the original text "...SFS must be heavy-tailed" was.

(6) Section A4 and Figure A1: can these calculations be compared to simulations?

We have more removed the previous A section on the deterministic analysis as they are not so relevant to our stochastic simulations indeed. Please see our response to reviewer 2 as well.

(7) Also, in general, please clarify how the results in the Appendix are used in the main text conclusions or provide insights relevant to these conclusions. If they are not, one can consider removing them.

We have more removed the previous A section on the deterministic analysis. The remaining sections are about stochastic simulations and extended figures which support our main figures.

(8) Figure A2: the two lines are difficult to tell apart on each panel. Please consider different styles.

We have changed one of the dotted lines to be solid. This figure is now Figure A1 in our revision.